# Biosynthesis of Gold Nanostructures and Their Virucidal Activity Against Influenza A Virus

**DOI:** 10.3390/ijms26051934

**Published:** 2025-02-24

**Authors:** Fernanda Contreras, Katherine Rivero, Jaime Andrés Rivas-Pardo, Fabiana Liendo, Rodrigo Segura, Nicole Neira, Mauricio Arenas-Salinas, Marcelo Cortez-San Martín, Felipe Arenas

**Affiliations:** 1Laboratorio de Microbiología Molecular, Facultad de Química y Biología, Universidad de Santiago de Chile, Santiago 9170022, Chile; fernanda.contreras.t@usach.cl (F.C.); katherine.rivero@usach.cl (K.R.); nicole.neira@usach.cl (N.N.); 2Laboratorio de Genómica Microbiana, Centro de Genómica y Bioinformática, Facultad de Ciencias, Ingeniería y Tecnología, Universidad Mayor, Santiago 8580745, Chile; jaime.rivas@umayor.cl; 3Laboratorio de Electroanálisis, Facultad de Química y Biología, Universidad de Santiago de Chile, Santiago 9170022, Chile; fabiana.liendo@usach.cl (F.L.); rodrigo.segura@usach.cl (R.S.); 4Centro de Bioinformática, Simulación y Modelado (CBSM), Facultad de Ingeniería, Universidad de Talca, Talca 3460000, Chile; marenas@utalca.cl; 5Laboratorio de Virología Molecular y Control de Patógenos, Facultad de Química y Biología, Universidad de Santiago de Chile, Santiago 9170022, Chile

**Keywords:** metal reduction, biosynthesis, gold nanostructures, virucidal, influenza A

## Abstract

Bacteria in natural environments often encounter high concentrations of metal ions, leading to the development of defense mechanisms such as chemical reduction. This process can result in the formation of nanostructures (NS) ranging from 1–100 nm, which have valuable properties for various applications, including as virucidal agents. Currently, metallic NS with virucidal activity are used in disinfectants and surface protection products. However, their production mainly relies on physical and chemical methods, which are often complex, toxic, and energy-intensive. A sustainable alternative is the biosynthesis of nanostructures. Our research focuses on the biosynthesis of gold nanostructures (AuNS) using environmental bacteria and their proteins, with the aim of exploring their potential as agents to destroy the influenza A virus. We screened bacteria under conditions with HAuCl_4_, identifying eight microorganisms capable of growing in high gold concentrations. *Staphylococcus haemolyticus* BNF01 showed the highest resistance and Au(III) reduction, growing up to 0.25 mM in HAuCl_4_. Bioinformatic analysis revealed five proteins with potential Au(III)-reductase activity, which were cloned and expressed in *Escherichia coli*. These proteins reduced gold to form AuNPs, which were purified, characterized for size, shape, and surface charge, and tested against influenza A, showing significant virucidal effects, likely due to interactions with viral proteins.

## 1. Introduction

The process of industrialization has caused significant ecological and environmental challenges, mainly due to the contamination of soils, sediments, and water bodies with metals. These pollutants are a major concern because metals accumulate in nature and do not degrade, leading to long-lasting effects on ecosystems and human health [1,2]. Microorganisms, as key components of terrestrial ecosystems, play a crucial role in nutrient cycles and energy transfer. However, they are highly sensitive to environmental changes, including metal pollution, which can disrupt their metabolic processes and community structures [3].

Metals vary in their biological role and toxicity. Some are essential for metabolic processes, while others are highly toxic even at low concentrations. Based on their toxicity, metals can be categorized as (i) essential and relatively non-toxic (e.g., calcium, magnesium); (ii) essential but toxic at high levels (e.g., iron, manganese, zinc); and (iii) highly toxic with no known biological function (e.g., mercury, cadmium, silver, and gold) [4]. The toxic effects of metals on microorganisms have been extensively studied, with mechanisms including the generation of reactive oxygen species (ROS), enzyme inhibition, and DNA damage. These toxic interactions disrupt redox balance, primarily through excessive ROS production, which leads to oxidative stress and cellular damage [5,6].

To survive in metal-contaminated environments, bacteria have evolved diverse resistance mechanisms, including metal efflux systems, extracellular sequestration, enzymatic detoxification, and biotransformation. One key detoxification strategy is the ability of certain bacteria to reduce metal ions into their elemental form, a process known as bioreduction, which can lead to the formation of metal nanostructures (NS) [7,8,9]. This microbial-driven metal transformation has gained significant attention as an eco-friendly alternative to traditional nanomaterial synthesis, which often relies on hazardous chemicals and energy-intensive processes.

Biogenic nanostructures, produced via bacterial enzymatic activity, offer multiple advantages over chemically synthesized counterparts. They are biocompatible, environmentally friendly, and free of toxic stabilizing agents, making them ideal for biomedical applications [10,11]. Among these, gold nanostructures (AuNS) have attracted interest due to their unique physicochemical properties, including their stability, tunable surface chemistry, and optical characteristics [12]. Studies have demonstrated the potential of AuNS as antimicrobial agents, capable of disrupting bacterial membranes and interfering with microbial metabolism. Additionally, AuNS have shown antiviral activity, particularly against enveloped viruses, including influenza A virus, by inhibiting viral replication and surface protein interactions [13]. However, the virucidal activity of metal nanoparticles, in general, has been poorly explored, and it remains an interesting area for future research.

Despite its promising applications, the microbial biosynthesis of AuNS remains poorly understood, particularly regarding the specific proteins and enzymatic mechanisms involved in Au(III) reduction. Most studies have focused on identifying bacterial species capable of metal bioreduction, but few have investigated the molecular basis of bacterial gold reduction and nanostructure formation. Understanding these biological processes is critical for optimizing green nanotechnology approaches and expanding the biomedical applications of biosynthesized nanomaterials [14,15].

This study aims to identify bacterial isolates with high gold resistance, characterize their ability to reduce Au(III), and investigate key reductase proteins involved in AuNS biosynthesis. By elucidating the microbial mechanisms responsible for gold nanostructure formation, this research seeks to provide a sustainable alternative for nanomaterial production with potential applications in biomedicine, particularly in the development of antiviral agents against influenza A virus.

## 2. Results

### 2.1. Evaluation of Resistance Levels and Reducing Capacity of Environmental Isolates Toward Au(III)

We evaluated the resistance of 18 environmental bacterial isolates to gold (HAuCl_4_) using minimum inhibitory concentration (MIC), minimum bactericidal concentration (MBC), and bacterial growth curves as proxy for the determination. To better understand the gold resistance potential among environmental bacterial isolates, we focused on six isolates that exhibited the highest Au(III) tolerance and reduction capacity, selecting them for further analysis. These isolates, originally from different sources in extreme environments such as Antarctica and Salars, exhibited quite different optimal growing temperatures ranging from 15 to 37 °C.

After completing the temperature characterization and optimization, we proceeded to explore the isolates’ ability to grow in the presence of gold. A comparative analysis of MIC and MBC values allowed the identification of the most resistant isolates, providing a basis for selecting the strains with the greatest potential for gold nanostructure biosynthesis. Table 1 summarizes the minimum inhibitory concentration (MIC) and minimum bactericidal concentration (MBC) values for gold for the environmental bacterial isolates. The highest resistance to gold was determined by comparing the MIC observed in each isolate with the control strain, *E. coli* BW25113. The MIC values for gold ranged from 0.25 mM to less than 0.0078125 mM, with isolates MF16, BNF01, BNF05, M53, and R37A showing the highest resistance. On the other hand, the highest resistance to gold was determined by comparing the MBC observed in each isolate with that of the control strain, *E. coli* BW25113. The MBC for gold ranged from 0.125 mM to 0.5 mM, with environmental isolates M53, M53B, and R37A demonstrating the highest resistance to the metal.

Based on these results, the environmental isolates that exhibited the highest resistance to gold compared to the *E. coli* BW25113 strain were selected for subsequent activities. The isolates MF01, MF16, BNF01, BNF05, BNF20, M53, M53B, and R37A were finally selected for further assays, evaluating growth curves in the presence of sublethal concentrations of gold. The growth parameters analyzed included maximum optical density at 600 nm (OD_600nm_), growth rate, and generation time (Table 1). Bacterial identification of the isolates selected was accomplished by sequencing the 16S rRNA gene using the universal primer 8F (forward) and 1492R (reverse). GeneBank accession numbers for the 16S rRNA nucleotide sequences are MG461635 (MF01), PV094902 (MF16), KF701037 (BNF01), KF701038 (BNF05), MF806171 (BNF20), PV101475 (M53), PV101473 (M53B), and PV096985 (R37A).

In Figure 1A,B, we show in detail the performance of BNF01 and MF16, environmental isolates with the best MIC results, in the presence of increasing gold concentrations. The environmental isolate MF16—*Staphylococcus* sp.—exhibited a growth similar to that of the control, although it was assayed at 62.5 µM of HAuCl_4_ (equivalent to 1/4 of the MIC) (Figure 1B). Notably, this environmental isolate did not show any variations in the growth parameters evaluated in the presence of gold salt (Table 1). The other six environmental isolates were also assayed, showing less capacity to grow in gold; however, based on the results obtained, we identified a total of six isolates resistant to gold: MF16, BNF01, BNF05, M53, M53B, and R37A. Moreover, some of these environmental isolates demonstrated resistance to more than one metal, such as isolates M53, M53B, and R37A, which showed resistance to both Au (III) and Zn (II). In particular, the isolate MF16 exhibited resistance to multiple metals, suggesting not only its ability to grow at high concentrations of toxic metal but also its resistance to different metals.

Later, we assayed the gold-reducing capacity of protein lysates from these six gold-resistant environmental bacterial isolates (MF16, BNF01, BNF05, M53, M53B, and R37A) at various pH from 6.0 to 9.0 in the presence of the cofactors NADH and NADPH (Figure 1C,D). In general, when assessing the gold-reducing activity of the crude extracts, we observed that most extracts exhibited activity across the majority of the pH levels tested in the presence of NADH, with the exception of the R37A extract. In contrast, activity in the presence of NADPH was observed only at specific pH levels (Figure 1D). The highest gold-reducing activity in the presence of NADH was detected at pH 8.0, for the protein lysate from BNF05 and MF16, while for NADPH, the MF16 protein extract at pH 7.4 shows the highest activity (Figure 1C,D).

In summary, we successfully identified a total of eight environmental isolates resistant to gold; however, only six showed outstanding performance at high gold concentrations: MF16, BNF01, BNF05, M53, M53B, and R37A. Additionally, the MF16 isolate displays the highest gold-reducing activity, suggesting that this isolate could be a strong candidate for subsequent in vivo or/and in vitro synthesis of gold nanostructures.

### 2.2. Identification, Amplification, and Cloning of Proteins with Potential Au(III) Reductase Activity in Environmental Isolates

To investigate the molecular basis of gold reduction in bacterial isolates, a targeted analysis was performed to identify key proteins associated with Au(III) reductase activity, prioritizing those with the highest predicted catalytic efficiency. We conducted a genome search to identify proteins with potential Au (III) reductase activity. We used the genome sequence of BNF01, which was available from previous studies [16].

After a literature search for proteins with known metal reductase activity, we performed an amino acid sequence analysis, yielding a dataset of 13,500 proteins classified by function and conserved motifs. We narrowed down the dataset to 63 proteins using the PDB, focusing only on proteins with structure available. Based on these results, five proteins with potential Au(III) reductase activity were predicted in the environmental isolate BNF01 (Table 2). Among the proteins identified, Gor exhibited the highest Au(III) reductase activity, making it the primary candidate for further biochemical characterization and gold nanostructure synthesis.

Following the in silico identification of proteins with potential Au(III) reductase activity, the genes encoding these proteins were amplified, expressed, and purified. Specific primers were then designed (Appendix A) to amplify the genes encoding proteins with probable Au(III) reductase activity from the genome of BNF01. Initially, PCR was performed using GoTaq DNA polymerase to confirm the functionality of the designed primers (Appendix A). Subsequently, high-fidelity Q5 DNA polymerase was used for PCR amplification of the target genes, and the products were visualized via agarose gel electrophoresis (Appendix A). PCR products of the expected molecular sizes were obtained for the four genes encoding proteins with potential Au(III) reductase activity from the genomic DNA of BNF01: 1524 bp for *cdr*, 724 bp for *oxi*, 1308 bp for *trmFO*, and 1323 bp for *gor* (Appendix A).

The PCR products of the expected sizes for each gene, obtained with Q5 DNA polymerase, were purified from the agarose gel for cloning into the pET101/D-TOPO and pET21b(+) vectors. The purified PCR products of the genes encoding proteins with probable Au(III) reductase activity were individually cloned into the pET101/D-TOPO (for the *cdr* gene) and pET21b(+) vectors. Recombinant vectors for each gene were used to transform into chemically competent *E. coli* TOP10 cells. After plating the bacteria on LB agar plates supplemented with ampicillin and streptomycin, we obtained transformant colonies containing the vectors pET101/*cdr*, pET21b/*gor*, pET21b/*oxi*, and pET21b/*trmFO*, which encodes the proteins with potential Au(III) reductase activity.

The introduction and correct orientation of the genes into the vectors were confirmed by colony PCR using either a set of specific primers to amplify the gene or a combination of a forward primer of the gene and a reverse primer only present downstream of the cloning site. The amplification yields the exact sizes of the *cdr*, *oxi*, *trmFO*, and *gor* genes, and for the orientation assay, 150 pb extra bp were founded to each amplicon, matching the extra segment belonging to the vector. Moreover, we used DNA sequencing to confirm the correct orientation in the plasmid and the absence of substitutions during the cloning process.

### 2.3. Expression, Purification, and Evaluation of Reductase Activity of Recombinant Proteins with Au(III) Reductase Activity

Recombinant expression and purification of the most promising Au(III) reductase candidates were performed to validate their enzymatic activity under controlled conditions and to optimize their use in gold nanostructure biosynthesis. The recombinant vectors containing the genes—pET101/*cdr*, pET21b/*gor*, pET21b/*oxi*, and pET21b/*trmFO*—were used to transform chemically competent *E. coli* BL21 (DE3) cells. After plating the bacteria on LB agar plates supplemented with ampicillin, transformant colonies were obtained for all cases. These transformed colonies were used to conduct induction assays at different temperatures (30 °C and 37 °C) and multiple time points (0, 1, 2, 4, 6, 8, and 24 h), aiming to determine the optimal expression conditions for each protein (Appendix A). In Figure 2, we show the purification of Gor protein, as an example of the recombinant protein expressed for the gold-reducing assays. After exploring the best conditions for the protein expression—assaying temperature and time after induction (Appendix A)—induction tests yield, in the case of Gor protein, a 48 kDa band when the bacteria is induced for 6 h at 30 °C. Similar conditions for induction were found for the other three enzymes, CoADR, TrmFO, and Oxi, which were also successfully purified.

Later, we purified the proteins through affinity chromatography, enriching the protein band belonging to the reducing enzyme. For Gor protein purification, the protein elutes in the first two fractions yielded a band above 48 kDa (Figure 2A, lanes 2–11). Subsequently, we pooled the fractions, evaluating the Au(III) reductase activity at multiple pH and in the presence of NADH or NADPH (Figure 2B,C).

The gold reductase activity assays indicate that all the enzymes, CoADR, TrmFO, Oxi, and Gor, display reducing activity across pH and in the presence of NADH and NADPH cofactors (Figure 2B,C). Nevertheless, Gor shows the highest gold reductase activity, registered at pH 9.0 in the presence of either NADH or NADPH, followed by TrmFO, which shows high reducing activity at pH 7.4 in the presence of NADPH (Figure 2C).

Based on the results obtained, the recombinant Gor protein was selected for the in vitro synthesis of gold nanostructures (AuNS), as it demonstrated the highest levels of reduction under the tested conditions, which could be related to the generation of a higher concentration of nanostructures.

### 2.4. Synthesis, Purification, and Characterization of Gold Nanostructures Generated In Vitro by Reductase Proteins

Gold nanostructures (AuNS) were synthesized in vitro using the purified recombinant protein from the environmental isolate BNF01. The optimal conditions for the reduction and synthesis of AuNS were determined to be 20 mM Tris-HCl buffer at pH 9.0, with 1 mM NADPH, 1 mM HAuCl_4_, and 50 μg of the purified recombinant Gor protein.

Following synthesis, the AuNS were purified and concentrated by centrifugation using 3 kDa filtration units (Amicon^®^, Darmstadt, Germany). The concentration of the biosynthesized AuNS was quantified using atomic absorption spectroscopy (AAS), yielding a concentration of 1149.9 ± 8.8 μg/mL.

The biosynthesized AuNS were then selected for further characterization using various techniques, as shown in Figure 3. Transmission electron microscopy (TEM) was used to determine the morphology and size of the AuNS. Different magnifications (Figure 3A) revealed that the AuNS exhibited some degree of aggregation and a homogeneous morphology, similar to the spiky or “urchin-like” shape described for this type of nanostructure. The sizes of the nanostructures varied, with a predominance of those larger than 50 nm.

The chemical composition of the AuNS was analyzed using scanning electron microscopy coupled with energy-dispersive X-ray spectroscopy (SEM–EDX) (ZEISS, Oberkochen, Germany; Oxford Instruments, Boston, MA, USA), revealing six peaks corresponding to carbon, oxygen, aluminum, and gold (Figure 3B). The AuNS had a gold content of 23%, with high levels of carbon (38.7%) and oxygen (38.0%), likely related to the biological synthesis of the nanostructures.

The UV–Vis absorbance spectrum of the AuNS showed a maximum absorbance at 622 nm, which is within the recorded range for nanostructures with an urchin-like morphology (555–702 nm) (Figure 3C). The size distribution and surface charge of the biosynthesized nanostructures were determined by dynamic light scattering (DLS) and zeta potential measurements (Figure 3D,E). The DLS analysis revealed two main size distributions of the AuNS, corresponding to 74 ± 17 nm and 6.7 ± 0.8 nm, with a polydispersity index (PDI) of 0.53 (Figure 3D). The zeta potential for the AuNS was −21.4 ± 4.05 mV, which suggests a tendency for the biosynthesized nanostructures to agglomerate (Figure 3E).

In summary, gold nanostructures (AuNS) were successfully synthesized using the recombinant Gor protein from the environmental isolate BNF01. These AuNS, already characterized to understand their physical, chemical, and optical properties, were purified, quantified, and stored for further analysis.

### 2.5. Propagation, Confirmation, and Quantification of Influenza A H1N1 Virus for Virucidal Assays

The influenza A H1N1 virus (ATCC No. VR-1736) was propagated in Madin–Darby canine kidney (MDCK) epithelial cell cultures to prepare a viral inoculum for subsequent virucidal assays in suspension and on non-porous surfaces. MDCK cell monolayers at 80–90% confluence were infected with a multiplicity of infection (MOI) of 0.1 of the influenza A H1N1 virus. The cytopathic effect (CPE) produced by the virus in cell culture was monitored daily (Appendix A). At 48 h post-infection, spaces began to appear in the MDCK cell monolayer (Appendix A), and by 72 h post-infection, nearly all cells had detached from the monolayer (Appendix A). This CPE contrasted with uninfected control cell cultures treated with 1X DPBS, where the cell monolayer remained intact at both 48 and 72 h post-infection (Appendix A). The supernatants from the infected cultures were collected and centrifuged to sediment cell debris, and the resulting supernatant was supplemented with 1% fetal bovine serum (FBS) and stored at −80 °C.

To confirm the presence of the influenza A H1N1 virus in the collected supernatants, we conducted an hemagglutination (HA) assay, based on the ability of influenza viruses to bind to red blood cells (Appendix A). Supernatants from two different infections (P0 and P1) were subjected to the hemagglutination assay, yielding 4 HA units for the P0 supernatant and 64 HA units for the P1 supernatant (Appendix A). In contrast, no hemagglutination was observed in the negative control (PBS) (Appendix A). These results confirmed that the post-infection supernatants contained influenza A virus with hemagglutinating activity.

The viral titers in the collected supernatants were then determined using two methods: the Reed–Muench method (TCID_50_/mL) and absolute RT-qPCR. To implement the RT-qPCR method, a recombinant plasmid containing a conserved segment of the influenza A M gene was generated to enable absolute quantification of the PCR product. Specific primers were used to amplify the conserved segment of the influenza A M gene from a cDNA synthesized by reverse transcription (RT) from RNA extracted from virus-containing supernatants. The amplification of the conserved segment of the influenza A M gene resulted in a PCR product of approximately 100 bp, consistent with the expected size of 101 bp (Appendix A). The PCR product was purified for subsequent cloning into the pGEM-T Easy vector.

The purified PCR product corresponding to the conserved segment of the influenza A M gene was cloned into the pGEM-T Easy vector. The resulting recombinant vector was used to transform chemically competent *E. coli* TOP10 cells. After plating the bacteria on LB agar plates supplemented with ampicillin and streptomycin, transformant colonies were obtained. These colonies were analyzed to confirm the correct insertion of the gene into the vector through colony PCR, restriction enzyme digestion, and sequencing (Appendix A). The colony PCR products confirmed the presence of the conserved M gene segment, showing a band of approximately 100 bp, consistent with the expected insert size (Appendix A). The gene’s cloning directionality in the vector was determined, revealing a band of over 200 bp, corresponding to the insert size plus 150 bp of the vector (Appendix A).

Recombinant vectors were purified from each of the verified colonies and digested with the restriction enzyme *Eco*RI. The digestion yielded two bands, one of approximately 3000 bp, matching the pGEM-T Easy vector size (3015 bp), and another of approximately 100 bp, corresponding to the insert size released from the vector (Appendix A). The correct insertion of the Mseg gene into the purified recombinant vectors was confirmed by sequencing, where the obtained sequence showed significant alignments with the influenza A virus matrix protein-coding segment 7 (Appendix A).

RT-qPCR amplification of 12 serial dilutions of known concentration of the purified recombinant vector pGEM-T Easy/Mseg was performed, yielding Ct values for each (Appendix A). From these data, the number of copies/µL of the conserved M gene segment in the recombinant vector was calculated (Appendix A).

Using RT-qPCR and the calibration curve obtained, the number of virus copies/mL in each of the collected viral supernatants was determined (Appendix A), with values ranging from 1.6 to 8.2 × 10^5^ copies/mL. However, these viral titers were insufficient for performing the virucidal assays in suspension and on non-porous surfaces, which require a minimum of 1.0 × 10^6^ copies/mL.

To obtain an inoculum with the necessary viral titers for the virucidal assays, the supernatant from influenza-virus-infected cells was concentrated using Amicon^®^ filtration units. RNA extracted from the concentrated viral supernatant was analyzed by RT-qPCR, resulting in a titer 100 times higher, equivalent to 1.43 × 10^7^ copies/mL in a final volume of 8 mL. This value was similar to that obtained using the Reed–Muench quantification method for TCID_50_/mL, where the concentrated viral supernatant had a viral titer of 5.6 × 10^7^ TCID_50_/mL.

### 2.6. Evaluation of the Virucidal Activity and Mechanism of Action of Biosynthesized AuNS Against Influenza A H1N1 Virus

The virucidal activity of biosynthesized gold nanostructures (AuNS) was evaluated against influenza A H1N1 virus both in suspension and on non-porous stainless-steel surfaces, following standardized contact times of 10 and 60 min as per the European Committee for Standardization [17,18]. A viral inoculum with a titer of 1.0 × 10^6^ TCID_50_/mL and a nanostructure concentration of 100 μg/mL was used in all tests. Standardized hard water (WSH) containing 300 ppm CaCO_3_ was used as a negative virucidal control, Virkon^®^, a peroxide-based disinfectant diluted 1:600, served as the positive control, and commercial CuNS < 10 nm were used as the biocidal nanostructure control.

In suspension assays, a reduction in initial viral titers was observed in the influenza A virus treatments with Virkon^®^ and biosynthesized AuNS at both contact times, and with commercial CuNS after 60 min of treatment (Figure 4A). On non-porous stainless-steel surfaces, a reduction in initial viral titers was observed following treatment with Virkon^®^ as well as with commercial CuNS after 60 min of treatment (Figure 4B). However, treatment with WSH and biosynthesized AuNS did not result in any changes in the initial viral titers at either contact time (Figure 4A,B).

Biosynthesized AuNS exhibited virucidal activity only in suspension assays, and showed virucidal efficacy at longer contact times, with a 4 log_10_ reduction in initial viral titers, comparable to the positive control (Virkon^®^) in both contact times, meeting the threshold for virucidal efficacy in surface assays (≥4 log_10_ reduction) (Table 3). In contrast, none of the tested nanostructures, including AuNS, were effective on non-porous surfaces, as no ≥ 3 log_10_ reduction in viral titers was observed (Table 3).

Following confirmation of the virucidal efficacy of AuNS after 60 min of treatment in suspension assays, various concentrations of these nanostructures (0–800 μg/mL) were used to determine the 50% inhibitory concentration (IC_50_) of the cytopathic effect in virucidal assays and the cytotoxicity of AuNS in MDCK cell cultures (Figure 4C). The results showed that the IC_50_ of AuNS was 77 ± 1 μg/mL (Figure 4C). Regarding cytotoxicity, it was observed that a concentration of 106 ± 1 μg/mL of AuNS resulted in 50% survival of the MDCK cell culture (Figure 4D).

To explore whether the virucidal action of AuNS was associated with the degradation of influenza A H1N1 surface proteins, a hemagglutination assay was conducted, with the virus treated for 60 min with nanostructures (Appendix A). The untreated virus (positive control) showed 16 HA units, while the PBS negative control did not exhibit hemagglutination in any dilution. The virus treated with WSH showed 2 HA units at the evaluated contact times, similar to treatments with commercial CuNS. In comparison, biosynthesized AuNS treatment resulted in 1 HA unit at 10 min and no HA units at 60 min. These results suggest that the hemagglutinin (HA) surface protein of the virus might be compromised after treatment with AuNS, as the hemagglutination capacity of influenza virus depends on the presence and integrity of this surface protein. However, these results do not confirm whether the viral particle itself has suffered structural damage, losing its integrity and, therefore, its hemagglutination capability.

To further investigate, influenza A viral particles were visualized by TEM after treatment with AuNS (Figure 5A,B). The electron micrographs of influenza A H1N1 virus treated with WSH for 60 min showed intact viral particles with an elongated spherical morphology, surrounded by a membrane with inserted surface proteins, with a diameter > 100 nm (average 144 ± 18 nm) (Figure 5A). In contrast, electron micrographs of viruses treated with AuNS for 60 min showed virus-like particles lacking surface proteins, with affected size (average 128 ± 21 nm) and irregular morphology, although maintaining their lipid membrane (Figure 5B). Additionally, electron-dense regions were observed, with an average size of 7.6 ± 1.5 nm, which could correspond to the smaller AuNS, potentially adsorbed on the virion surface (Figure 5B).

To confirm whether the smaller AuNS could be adsorbed on the surface of influenza A H1N1 virions, we conducted a gold quantification after the treatment of nanostructures using AAS. The virus–AuNS mixture was incubated for 60 min at a gold concentration of 107 μg/mL. When the treated virus samples were centrifuged after 60 min of incubation, the sediment had a gold concentration of 86.7 μg/mL, while the supernatant had 13.6 μg/mL of gold. In contrast, the AuNS control after centrifugation showed a gold concentration of 103.2 μg/mL in the sediment, with no gold detected in the supernatant. No gold was detected in the sediment or supernatant of the untreated virus control after centrifugation. These results suggest that approximately 13.6% of the AuNS interacted with the influenza A virion after treatment, likely corresponding to the smaller nanostructures observed by TEM and DLS (Figure 3A,D).

The effect of AuNS treatment on virion proteins was further evaluated through SDS-PAGE analysis (Figure 5C). Influenza A H1N1 virus treated with WSH for 60 min showed an intense band pattern corresponding to viral proteins (Figure 5C, lanes 2–4). However, after treatment of the enriched influenza virus aliquot with AuNS for 60 min, a decrease in band intensity was observed, possibly associated with the degradation of some viral proteins (Figure 5C, lanes 5–7).

In summary, the biosynthesized AuNS exhibited virucidal action against influenza A virus in suspension assays, likely related to the degradation of viral proteins, although this effect could not be associated with any specific protein in the influenza A virion. Further investigations would be required to elucidate the precise mechanisms underlying this virucidal activity.

## 3. Discussion

### 3.1. Resistance Levels and Reducing Capacity of Environmental Isolates Toward Au(III)

The minimum inhibitory concentration (MIC), defined as the lowest concentration of a compound that completely inhibits the growth of a microorganism, has been widely used to identify bacterial resistance and susceptibility to specific compounds [19]. However, there are limited studies concerning the toxic effects of tetrachloroauric acid on bacteria. For instance, we previously reported in Muñoz-Villagrán et al. (2020) [20] an MIC of 0.25 mM for tetrachloroauric acid in *E. coli* under aerobic conditions, which is higher than the 0.125 mM MIC observed for the same strain in this study. The highest MIC observed in this work for tetrachloroauric acid was 0.25 mM, with isolates MF16, BNF01, BNF05, M53, and R37A showing the highest resistance.

The minimum bactericidal concentration (MBC), defined as the minimum concentration of a compound that results in bacterial death [21], was also evaluated. For *E. coli* BW25113, the MBC of tetrachloroauric acid was found to be 0.25 mM, while the most resistant environmental isolates (M53, M53B, and R37A) exhibited an MBC of 0.5 mM. These results indicate that gold is particularly toxic to the environmental isolates, requiring low concentrations to inhibit growth and exert bactericidal effects. This high level of toxicity could be attributed to the non-essential nature of gold for cellular processes and the limited resistance mechanisms in bacteria against the oxidative stress induced by gold ions, which include intracellular thiol depletion and superoxide anion accumulation, leading to damage in DNA, proteins, and membranes [20].

In parallel, the metal-reducing activity of the environmental isolates was evaluated using crude bacterial extracts. Previous research has linked the metal-reducing activity exhibited by bacterial crude extracts with the cell’s ability to reduce metals [16,22]. In this study, the reducing activity toward Au(III) was assessed at 37 °C across a pH range of 6.0 to 9.0, in the presence of either NADH or NADPH, to determine the isolates’ capacity to reduce this metal. Although a correlation was observed between the resistance levels of the isolates and their ability to reduce gold, this reduction was not identified as the primary mechanism of resistance.

The crude extracts displayed gold-reducing activity under various conditions, with activity observed at all tested pH levels and in the presence of both NADH and NADPH. The primary component of the crude extracts, given the extraction procedure, consists of the proteins within the microorganisms [23]. Thus, the metal-reducing activity observed is likely mediated primarily by these protein components. It has been reported that pH influences metal speciation, allowing the formation of complexes or the protonation/deprotonation of functional groups present in amino acids, which participate in stabilizing the enzyme–substrate complex, explaining the activity observed across the tested pH range [24].

Moreover, other molecules within the crude extracts, such as nucleic acids and polysaccharides, could also mediate the reduction of Au(III). Polysaccharides, under alkaline conditions, are capable of forming highly reactive molecules (such as enediols), which may contribute to the reduction of gold ions [25]. The preference for either NADH or NADPH as cofactors could be due to their stability within the active site of the proteins involved in the reduction process [26].

Overall, the study demonstrated that while environmental isolates showed resistance to gold and exhibited reducing activity under various conditions, this reduction activity was not the primary mechanism of resistance, suggesting that these bacteria might possess other mechanisms to cope with gold toxicity.

### 3.2. Identification, Cloning, and Reductase Activity of Proteins with Probable Au(III) Reductase Activity

In this study, bioinformatic approaches were employed to identify proteins with potential Au(III) reductase activity, crucial for understanding the mechanisms by which bacteria might reduce gold ions. The first approach involved an extensive literature review to identify proteins previously reported to have metal reductase activity. Amino acid sequences of these proteins were analyzed using various databases, including PubMed, Uniprot, and Protein Data Bank, to find conserved motifs and domains that might indicate reductase activity. The second approach focused on a structural analysis of proteins known for their metal reductase functions, applying specific filters to uncover new proteins with similar activities. These methods collectively led to the identification of five proteins in *Staphylococcus haemolyticus* (BNF01) with probable Au (III) reductase activity, expanding our understanding of bacterial adaptation to toxic metal environments.

These proteins, identified primarily as oxidoreductases, catalyze redox reactions involving the transfer of electrons from a donor molecule to an acceptor molecule, facilitating the reduction of metal ions like Au(III). For example, glutathione reductase, typically involved in reducing glutathione disulfide to glutathione, was also predicted to possess Au(III) reductase activity, suggesting a secondary function beyond its primary enzymatic role [27]. Similarly, methylenetetrahydrofolate-tRNA-(uracil-5)-methyltransferase (TrmFO), which catalyzes the formation of 5-methyluridine in tRNAs, might reduce Au(III) as a secondary function [28].

The bioinformatic search used advanced servers and tools like InterProScan, Pfam, Prosite, Superfamily, and CATH to identify conserved domains and motifs. This strategy has been previously employed to identify six proteins with probable tellurite reductase activity in *E. coli*, including TrxB, AhpF, YkgC, GorA, E3, and NorW [29]. Of the 16 proteins identified in this study, most were classified as oxidoreductases, except for TrmFO, classified as a transferase. The ability of oxidoreductases to facilitate electron transfer plays a crucial role in reducing metal ions like Au(III), potentially leading to the formation of nanostructures. Moreover, the identified proteins, particularly flavoproteins containing a flavin adenine dinucleotide (FAD) prosthetic group, are known for their role in metal and metalloid reduction [30]. Notably, flavoproteins like alkyl hydroperoxide reductase, thioredoxin reductase, and flavorubredoxin reductase have shown reductase activity towards tellurite, while glutathione reductase has demonstrated activity against tellurite, selenite, and gold [16,31].

The identified genes encoding these proteins were cloned into expression vectors, with some challenges encountered during the process. Specifically, the *trxB* gene did not yield transformant colonies, likely due to an inappropriate insert-to-vector ratio or potential damage during gel purification [32]. Despite this, other genes were successfully cloned into pET101/D-TOPO and pET21b(+) vectors, allowing for the production of recombinant proteins under the control of the T7 promoter. This promoter, inducible by IPTG, facilitates high levels of protein expression in *E. coli* BL21 (DE3) cells. However, induction assays revealed that the optimal expression conditions varied across proteins, with some, like Oxi and TrmFO, showing low expression levels. This may be attributed to rare codons in the sequences, affecting translation efficiency, or high GC content at the 5′ end of the mRNA, impacting stability and translation [33,34].

Purification of these recombinant proteins was conducted using Ni^+2^-sepharose affinity chromatography, leveraging a 6xHis tag for selective binding. While CoADR and Gor were purified successfully with single bands representing the expected molecular weight, proteins like Oxi and TrmFO exhibited multiple bands, possibly due to contaminants or partial protein degradation. Such issues underscore the challenges in producing pure recombinant proteins, particularly those with metal reductase activity, necessitating further purification steps, including concentration through filtration units.

The reductase activity of these purified proteins was then assessed under varying pH conditions (6.0 to 9.0) and in the presence of NADH or NADPH, aiming to identify the optimal conditions for Au(III) reduction. The activity levels observed in these assays were lower than those recorded using crude extracts, suggesting that additional cellular components might contribute to metal reduction, as previously discussed. Notably, the proteins CoADR, TrmFO, Oxi, and Gor showed significant gold reductase activity, particularly at pH 9.0 with NADPH for Gor and at pH 7.4 in Tris-HCl buffer with NADPH for TrmFO. These findings align with previous studies reporting similar pH preferences for Au(III) reductase activity in other proteins, such as sulfite reductase and lignin peroxidase [14,35].

The reductase activity observed at pH levels of ≥7.4 likely relates to the basic conditions favoring the deprotonation of cysteine residues in the catalytic sites, leading to highly reactive thiolate anions, crucial for the reduction of Au(III) [36]. The study’s findings highlight the potential of these proteins, particularly under specific pH conditions, to facilitate gold reduction, with implications for nanostructure formation and other biotechnological applications.

### 3.3. Synthesis, Purification, Quantification, and Characterization of AuNS Synthesized In Vitro by the Gor Reductase Protein

Gold nanostructures (AuNS) were synthesized in vitro using the purified recombinant Gor protein from the environmental isolate *Staphylococcus haemolyticus* (BNF01). The synthesis was conducted under optimal conditions: 20 mM Tris-HCl buffer at pH 9.0, 1 mM NADPH, 1 mM HAuCl_4_, and 50 µg of Gor protein. The formation of AuNS was visually confirmed by a color change in the reaction mixture from pale yellow to dark purple after 24 h at 37 °C, consistent with the synthesis of AuNS [37]. The generation of AuNS through biological methods, particularly using proteins, has been documented in various studies. For example, the recombinant gold reductase (GoIR) from *Erwinia* sp. IMH was shown to generate AuNS when incubated with NADH and HAuCl_4_ in Tris-HCl at 30 °C [38]. Similarly, a NADPH-dependent sulfite reductase from *E. coli* produced AuNS at 25 °C when mixed with HAuCl_4_ and other components in phosphate buffer [14]. The use of glutathione reductase for AuNS synthesis has also been reported, where the enzyme was incubated with NADPH and HAuCl_4_, leading to electron transfer from NADPH to the FAD moiety of the enzyme, ultimately reducing AuCl^4−^ to Au^0^ and stabilizing the nanostructures [31]. The direct role of NADH/NADPH cofactors alone in AuNS synthesis (abiotic synthesis) was not evaluated in this research, as the study was focused on the enzymatic generation of nanoparticles. However, controls of the nanoparticle synthesis reaction were performed without the Gor protein, conditions under which no abiotic AuNS formation was obtained. Despite this, under certain conditions of pH, temperature, and ionic strength, NADH/NADPH cofactors could contribute to metal reduction, and this remains an interesting aspect for future research.

The biosynthesized AuNS were purified and concentrated by centrifugation using 3 kDa filtration units (Amicon^®^), which effectively removed proteins and other reaction components that might interfere with subsequent applications. The quantification of AuNS was performed using atomic absorption spectroscopy (AAS), which determined the concentration of gold in the samples, yielding 1150 μg/mL. AAS is a widely used technique for quantifying metal concentrations in samples, providing accurate and sensitive measurements [39].

Characterization of the AuNS was conducted using several analytical techniques to ensure that they were within the nanoscale range and to explore their properties. Transmission electron microscopy (TEM) (Hitachi High-Technologies Corporation, Tokyo, Japan) revealed that the AuNS exhibited a homogeneous morphology, predominantly in the shape of branched or spiky structures, often referred to as “urchin-like” gold nanostructures. Such morphologies have been described in previous studies and are known for their unique optical properties and broad absorption spectra [40]. The synthesized AuNS varied in size, with most falling within the 10–20 nm and 70–80 nm ranges, though some aggregation was observed, likely due to the sample preparation method for TEM analysis.

Scanning electron microscopy coupled with energy-dispersive X-ray spectroscopy (SEM–EDX) confirmed the chemical composition of the AuNS, which contained 23% gold, along with significant amounts of carbon (38.7%) and oxygen (38.0%). The presence of carbon and oxygen is likely associated with stabilizing agents, such as amino acids or other biomolecules from the Gor protein, which were not fully removed during purification [41]. These stabilizers play a crucial role in determining the surface chemistry, morphology, and size distribution of the nanostructures. UV–Vis spectroscopy revealed that the biosynthesized AuNS had a maximum absorbance at 622 nm, consistent with the absorbance range for urchin-like nanostructures, which typically exhibit broad absorption properties due to their complex morphology and size distribution [42]. The broad peak observed in the UV–Vis spectrum corresponds to the polydispersity of the AuNS, which was further supported by dynamic light scattering (DLS) analysis. DLS measurements showed two predominant size distributions at 6.68 nm and 74.38 nm, with the smaller particles being more abundant. However, these results do not agree with those obtained through TEM, which is mainly due to sample preparation: (i) for TEM imaging, multiple droplets were deposited on the copper grid, leading to increased local concentrations and aggregation of nanoparticles due to their zeta potential; (ii) for DLS analysis, the sample was sonicated prior to measurement, which prevented aggregation and reflected the true size distribution of individual particles.

The zeta potential of the AuNS was measured as −21.4 mV, indicating that the nanostructures possess moderate stability with a tendency to aggregate. Zeta potential values within the range of −10 to +10 mV are typically associated with neutral particles, while values exceeding ±30 mV indicate strong repulsive forces and greater stability [43]. Regarding interaction with biological systems, it has been described that nanostructures with positive zeta potential tend to bind to negatively charged surfaces and/or molecules, and vice versa, so the biosynthesized AuNS could interact with components that have a positive charge, such as the hemagglutinin protein of the influenza A virus [44,45]. This potential interaction highlights the importance of zeta potential in the stability and biological interactions of AuNS.

Overall, the successful synthesis, purification, quantification, and characterization of AuNS using the Gor protein demonstrate the potential of biological methods for producing nanomaterials with specific properties tailored for various applications. This study adds to the growing body of research on the biosynthesis of metal nanostructures, highlighting the advantages of using recombinant proteins for controlled and efficient nanoparticle production.

### 3.4. Propagation and Quantification of Influenza A H1N1 Virus in Cell Culture

The propagation of influenza A H1N1 virus in MDCK cell cultures was essential for producing viral inoculum for virucidal assays in suspension and on non-porous stainless-steel surfaces. MDCK cells were selected due to their documented high susceptibility to influenza A virus, which often results in higher viral titers compared to other cell lines such as A549 or Vero [46]. Upon infection, MDCK cells exhibited a typical cytopathic effect (CPE), including significant cell detachment and loss of monolayer integrity 48 to 72 h post-infection, consistent with established findings [47,48].

To verify the presence of the virus in the collected supernatants, a hemagglutination (HA) assay was performed. This assay leverages the hemagglutinin protein’s ability to bind to red blood cells, confirming successful propagation with HA units of 4 and 64. While HA assays can estimate viral quantities, it is important to recognize that various factors, such as erythrocyte quality and ambient temperature, can influence results [49].

For more precise quantification, a recombinant plasmid containing a conserved segment of the matrix (M) protein gene was constructed to serve as a calibration standard for RT-qPCR. This approach, based on methods developed by Spackman et al. (2002) [50] and Furuse et al. (2009) [51], facilitated the absolute quantification of viral RNA. The analysis revealed viral titers ranging from 1.6 to 8.2 × 10^5^ copies/mL, insufficient for virucidal assays that require at least 1 × 10^6^ copies/mL [52].

To address this issue, viral supernatants were concentrated using Amicon^®^ ultrafiltration units, a method selected for its ability to concentrate viral particles while maintaining infectivity. This procedure resulted in a 100-fold concentration, yielding final viral titers of 1.43 × 10^7^ copies/mL by RT-qPCR and 5.6 × 10^7^ TCID_50_/mL by Reed–Muench assay, within the expected range for such concentration techniques [53]. The success of this approach highlights the significance of reliable concentration methods in achieving adequate viral titers for antiviral testing, especially when initial propagation yields are suboptimal. Additionally, the potential use of alternative MDCK variants like MDCK-SIAT1 or MDCK-London, which may produce higher viral titers due to differences in sialic acid receptor expression, could be considered for future studies [54,55].

### 3.5. Virucidal Activity of Biosynthesized AuNS Against Influenza A H1N1 Virus

The virucidal activity of the biosynthesized gold nanostructures (AuNS) and commercial copper nanostructures (CuNS) against influenza A H1N1 virus was evaluated through suspension assays and on non-porous stainless-steel surfaces. The results showed a reduction in viral titers with Virkon^®^ and biosynthesized AuNS in both short (10 min) and long (60 min) contact times in suspension assays, while commercial CuNS only reduced viral titers after 60 min. On stainless steel surfaces, viral titer reduction was observed with Virkon^®^ at both contact times and with commercial CuNS after 60 min.

The expected reduction in the viral titers with Virkon^®^ confirms its effectiveness as a positive control, given its established efficacy against influenza A virus [56]. The commercial CuNS, with a hydrodynamic diameter of <10 nm, are known biocidal agents, but the specific concentration and contact time required for virucidal activity were not provided, likely necessitating higher concentrations for shorter contact times.

Biosynthesized AuNS demonstrated virucidal activity against influenza A H1N1 in both short and long contact times during suspension assays, effectively reducing viral titers. Previous studies have shown that porous and non-porous AuNS can inactivate the influenza A virus, achieving nearly 100% cell viability at 0.2 mg/mL and 50% at 0.1 mg/mL, indicating viral inactivation and cell survival [57]. Comparatively, the biosynthesized AuNS in this study exhibited stronger virucidal effects at 100 µg/mL, suggesting enhanced efficacy at lower concentrations.

Additionally, Chaika et al. (2022) [58] reported that 5 nm and 20 nm AuNS synthesized chemically via sodium citrate reduction exhibited 80% and 50% inhibition of viral infection, respectively, at 1.97 μg/mL. However, these nanostructures were cytotoxic at the tested concentration, contrasting with the lower toxicity of biosynthesized AuNS.

Virucidal efficacy is defined as a reduction of ≥4 log_10_ in viral titers in suspension assays or ≥3 log_10_ on non-porous surfaces, equating to a 99.9% reduction in microorganisms [17,18]. Based on this definition, only biosynthesized AuNS demonstrated effective virucidal activity against influenza A H1N1 during long contact times in suspension assays, achieving a 4 log_10_ reduction in viral titers. However, to confirm general virucidal efficacy, these gold nanostructures should be tested against three non-enveloped viruses—poliovirus type 1, adenovirus type 5, and murine norovirus—as these are highly resistant to various agents, and effectiveness against them would indicate broader antiviral potential [17,18].

### 3.6. Inhibitory Concentration (IC_50_) and Cytotoxicity of Biosynthesized AuNS

The biosynthesized gold nanostructures (AuNS) were found to be effective viricidal agents against influenza A H1N1 virus after 60 min of treatment in suspension assays, with an IC_50_ determined to be 76.64 ± 1.08 μg/mL. In comparison, a study by Kim et al. (2020) [57] observed that treatment with non-porous spherical AuNS at a concentration of 0.2 mg/mL and porous AuNS at 0.1 mg/mL led to a 50% inhibition of infection in MDCK cell cultures after 60 min, indicating that biosynthesized AuNS in this study are more effective at lower concentrations [57]. Similarly, another study reported an IC_50_ of 210 µg/mL using AuNS biosynthesized with *Glaucium flavum* leaf extract [59], further highlighting the relative efficiency of the biosynthesized AuNS.

Additionally, the cytotoxicity of the biosynthesized AuNS was evaluated in MDCK cell cultures over 48 h, revealing a 50% cell survival rate at an exposure concentration of 106.3 ± 1.2 μg/mL. Other studies have assessed the cytotoxicity of AuNS of different sizes, showing 59% cell survival after exposure to 1.97 μg/mL of 5 nm AuNS and 76% survival with 1.97 μg/mL of 20 nm AuNS [58]. Despite the concentration used in these viricidal assays being nearly 100 times higher, the 5 nm and 20 nm AuNS exhibited greater toxicity compared to the biosynthesized AuNS. Differences in toxicity could be attributed to various characteristics of the nanostructures, such as shape, size, surface charge, concentration, and exposure time [60].

Although the biosynthesized AuNS are effective viricidal agents at 100 μg/mL, and exhibit a 50% cell survival rate at this concentration, additional studies are necessary to evaluate their potential as disinfectants or surface protectants. These studies should include further toxicity assessments using in vitro models (e.g., organoids) or in vivo studies (e.g., small animal models) to ensure that they pose no risk to health or the environment [61].

### 3.7. Probable Mechanism of Virucidal Action of Biosynthesized AuNS Against Influenza A H1N1 Virus

To elucidate the potential mechanism through which biosynthesized gold nanostructures (AuNS) exert their virucidal effect against the influenza A H1N1 virus, we first evaluated the possible damage to the viral RNA genome after treatment with AuNS for 10 and 60 min using RT-qPCR. The results indicated that treatment with Virkon^®^ and commercial CuNS did not lead to significant changes in the initial viral M gene copies, which is consistent with the known action of Virkon^®^, a disinfectant that disrupts cellular membranes through peroxygen-based compounds [62]. Similarly, CuNS are thought to inactivate viruses by releasing Cu^+1^ and Cu^+2^ ions, generating highly reactive hydroxyl radicals that can damage cellular components, although these radicals have a very short half-life, potentially explaining the lack of observed viral RNA degradation [63,64]. In contrast, AuNS-treated samples showed a slight decrease in viral RNA copies, suggesting some genomic damage, but the reduction was minor (<1 log_10_), indicating that this may not be the primary mechanism of virucidal action.

A more in-depth investigation focused on whether AuNS could cause degradation of viral proteins, particularly hemagglutinin, a critical protein for the virus’s ability to attach to host cells. Hemagglutination (HA) assays revealed that after 10 min of treatment with biosynthesized AuNS, the HA units decreased from 2 to 1, and after 60 min, hemagglutination was completely inhibited. This reduction suggests that hemagglutinin’s structural integrity may be compromised by AuNS, as hemagglutination relies on the intact function of this protein to bind red blood cells [65]. In contrast, treatment with CuNS, both biosynthesized and commercial, did not reduce HA units, indicating that these nanostructures did not affect hemagglutinin.

To further assess the impact of AuNS on viral particles, transmission electron microscopy (TEM) was employed. Untreated influenza A virions exhibited the expected morphology with a well-defined lipid membrane and surface proteins, consistent with known characteristics of influenza virions [66]. However, AuNS-treated virions displayed significant alterations, including loss of surface proteins, reduction in average diameter from 144 nm to 128 nm, and irregular shapes, although the lipid membrane remained intact. These observations align with studies in which nanostructures were shown to adsorb onto viruses, causing inactivation without completely destroying the virions [57,67]. Additionally, the presence of electron-dense regions within the treated virions, likely corresponding to smaller AuNS adsorbed on the viral surface, was supported by atomic absorption spectroscopy (EAA), which quantified approximately 13.6 µg/mL of AuNS associated with the virions.

Further examination of viral proteins through SDS-PAGE revealed a decrease in the intensity of viral protein bands after AuNS treatment, suggesting that the nanostructures might cause protein degradation. This result is consistent with findings from studies involving CuNS, in which an increase in nanostructure concentration correlated with reduced protein band intensity, indicating protein degradation [68]. The additional bands observed in the SDS-PAGE analysis could be attributed to cellular proteins incorporated into virions during assembly, a phenomenon previously noted in influenza virus assembly [69]. These findings suggest that protein degradation may play a key role in the virucidal mechanism of biosynthesized AuNS. However, further experiments, such as Western blotting with specific antibodies, are necessary to confirm which viral proteins are most affected by AuNS treatment.

Spectrophotometric measurements at 280 nm were conducted to detect potential degradation products of viral proteins. The results showed no significant change in absorbance over the 60 min treatment with AuNS, which might be explained by the generation of smaller peptides that continue to absorb at this wavelength despite protein degradation [70]. Given the known affinity of AuNS for sulfhydryl groups, the Ellman assay was used to determine whether the AuNS affected these groups within viral proteins. The assay results indicated no significant reduction in sulfhydryl content, suggesting that AuNS did not specifically interact with cysteine residues in hemagglutinin or neuraminidase, despite these proteins having multiple cysteine residues [71]. Previous studies have shown that porous AuNS, which have a larger surface area than non-porous AuNS, are more effective in interacting with viral proteins, potentially explaining the differences in interaction observed in this study [57].

In summary, while biosynthesized AuNS appear to exert their virucidal action through multiple mechanisms, including potential minor damage to viral RNA and significant protein degradation, particularly of hemagglutinin, the primary mode of action likely involves structural disruption of the virus, leading to loss of infectivity. The findings suggest that AuNS could serve as potent antiviral agents, although further research is needed to fully understand their interactions with viral components and to confirm their efficacy and safety in broader applications.

## 4. Materials and Methods

### 4.1. Bacterial Strains and Growth Conditions

Environmental bacterial isolates from Chile, Peru, and Bolivia (Appendix A) were cultured in Luria Bertani (LB) medium (10 g/L tryptone, 10 g/L NaCl, 5 g/L yeast extract) at their optimal growth temperatures, previously determined in our lab. We used LB-agar (2% *w/v*), with incubation overnight at the respective optimal temperatures, identifying the multiple strains, including *Enterobacter cloacae* (MF01), *Staphylococcus* sp. (MF16), *Staphylococcus haemolyticus* (BNF01), *Staphylococcus sciuri* (BNF05), *Psychrobacter immobilis* (BNF20), *Pseudomonas weihenstephanensis* (M53, R37A), and *Pseudomonas arsenicoxydans* (M53B).

On the other hand, *Escherichia coli* strains (Appendix A) used in the recombinant protein expression assays were grown in LB medium at 37 °C with shaking (150 rpm), or solid medium of LB-agar (2% *w*/*v*) was used, incubated overnight at 37 °C. *E. coli* TOP10 cells (Invitrogen^®^, Waltham, MA, USA) transformed with the DNA vectors (pET101/D-TOPO (Invitrogen^®^), pET21b(+) (Novagen^®^, Darmstadt, Germany), and pGEM-T Easy (Promega^®^, Madison, WI, USA), were cultured with 100 µg/mL ampicillin and 50 µg/mL streptomycin, while *E. coli* BL21(DE3) cells (Invitrogen^®^) were cultured with 100 µg/mL ampicillin.

### 4.2. Minimum Inhibitory Concentration (MIC), Minimum Bactericidal Concentration (MBC), and Bacterial Growth Parameters

MIC and MBC were determined following Benhalima et al. (2019) [72]. Serial dilutions of HAuCl_4_ (Sigma-Aldrich, Burlington, MA, USA) starting at 1 mM were prepared, and 10 µL of bacterial cultures (OD_600nm_ of 0.5) were added to each well in a 48-well plate. After 24 h of incubation with shaking at the optimal temperature, MIC was identified by turbidity, while MBC was confirmed by plating 100 µL from non-turbid wells on LB agar and observing the absence of colonies after 24 h.

For bacterial growth curves, overnight cultures of environmental isolates were diluted and incubated in fresh LB medium containing sublethal metal concentrations. Growth was monitored at 600 nm every 30 min for 18 h using a TECAN Infinite^®^ M200 Pro plate reader (Mannedorf, Switzerland). Growth parameters, including maximum OD_600nm_, growth rate (µ), and generation time, were calculated from the logarithmic growth phase and compared with control conditions.

### 4.3. Preparation of Crude Bacterial Extracts

Crude bacterial extracts were prepared by centrifuging cultures grown to an OD_600nm_ of 0.5 at 9000 rpm for 10 min at 4 °C. The pellet was resuspended in 20 mM Tris-HCl buffer (pH 7.4) with 0.1 mM PMSF (Roche^®^, Basel, Switzerland), then sonicated on ice (3 cycles of 5 min, 60% amplitude). The resulting solution was centrifuged at 14,000 rpm for 10 min at 4 °C, and the supernatant was collected as the crude extract. Protein concentrations were determined using the Bradford method (Sigma-Aldrich) with BSA (Sigma-Aldrich) as a standard, measuring absorbance at 595 nm [73]. Additionally, protein quantification in chromatography fractions and treated virions was done at 280 nm using a TECAN Infinite^®^ M200 Pro plate reader.

### 4.4. Gold-Reducing Activity Mediated by Crude Extracts

The metal-reducing activity of Au(III) contained in the respective salts HAuCl_4_ exhibited by crude bacterial extracts from environmental isolates was evaluated, as described by Figueroa et al. (2018) [16], at the optimal growth temperature for each isolate in a range of pH values, using 20 mM potassium phosphate buffer at pH 6.0, 7.0, and 7.4, and 20 mM Tris-HCl buffer at pH 7.4, 8.0, and 9.0. The reaction was performed in a volume of 200 µL in the corresponding pH buffer containing 50 µg of crude extract from the isolate, 1 mM NAD(P)H (NADH Sigma-Aldrich; NADPH Roche^®^), and the metal salt to be evaluated at a concentration of 1 mM. The production of metal in its elemental state was monitored spectrophotometrically in a range of 400–800 nm for gold. One enzyme unit (U) was defined as the amount of enzyme required to increase the absorbance by 0.001 units per minute under assay conditions. Enzyme activity was normalized by protein concentration.

### 4.5. Identification of Proteins with Potential Gold-Reducing Activity

An in silico search for proteins with potential metal-reducing activity was conducted through amino acid sequence and structural analysis at the Center for Bioinformatics and Molecular Simulation at the University of Talca. For sequence analysis, a literature search was carried out using PubMed and Oxford Journal platforms, with sequences obtained from the Uniprot platform. The sequences were analyzed using InterProScan (accessed on 20 April 2023), along with Pfam (accessed on 29 May 2023), ProSite (accessed on 3 June 2023), and Superfamily (accessed on 14 June 2023) databases to identify conserved patterns and motifs [74,75,76,77,78]. For structural analysis, known metal-reducing proteins such as thioredoxin reductase, dihydrolipoamide dehydrogenase, and oxidoreductase were used. The Protein Data Bank (PDB) platform was utilized to find new proteins that bind metals of interest (gold), contain FAD, and share 70% identity. Identified proteins were classified using CATH, RMSD, and SSAP servers [29]. These proteins were then subjected to a BLASTp search against available amino acid sequences from environmental isolates that showed the highest resistance and reduction according to assays performed. Once identified, the nucleotide sequences were obtained from the NCBI database using the protein names from the BLASTp search.

### 4.6. Cloning and Molecular Biology Techniques

Genomic DNA from environmental isolate BNF01 was extracted using the Wizard^®^ Genomic DNA Purification Kit (Promega^®^, Madison, WI, USA), with integrity verified via agarose gel electrophoresis and quantification performed as required. We designed specific primers (Appendix A) using Snapgene software 8.0 (GSL Biotech, San Diego, CA, USA), which were synthesized by Integrated DNA Technologies (IDT, Coralville, IA, USA), for the amplification of genes with potential metal-reducing activity, such as *trxB*, *cdr*, *oxi*, *trmFO*, and *gor*. These primers were tailored for use with the pET101/D-TOPO and pET21b(+) vectors, ensuring correct insertion sites and directional cloning.

The amplification of target genes was carried out using PCR with high-fidelity DNA polymerases like Q5 (NEB^®^, Ipswich, MA, USA) and Gotaq (Promega^®^). Construct preparation involved digesting the pET21b(+) vector and PCR products with *Nhe*I-HF and *Hind*III-HF enzymes (NEB^®^) for precise cloning. For pET101/D-TOPO cloning, the *cdr* gene from BNF01 was utilized, while the conserved segment of the influenza A virus matrix protein gene (*Mseg*) was cloned into the pGEM-T Easy vector. These constructs were verified by colony PCR, ensuring the correct orientation and presence of inserts.

Following transformation of *E. coli* TOP10 and BL21(DE3) cells, recombinant plasmids were purified using the FavorPrep™ Plasmid DNA Extraction Mini Kit (Favorgen^®^, Taiwan, China). The constructs generated included pET101/*cdr*, pET21b/*gor*, pET21b/*oxi*, pET21b/*trmFO*, pET21b/*trxB*, as well as pGEM-T Easy/*Mseg* from influenza A virus. Each construct was sequenced in full using Oxford Nanopore Technologies (ONT) long-read sequencing (Plasmidsaurus Inc., Monrovia, CA, USA), with sequence data analyzed via Snapgene and BLASTn databases to confirm accuracy.

Agarose gel electrophoresis (1% or 2% *w*/*v*) was routinely used to assess the integrity of genomic DNA, plasmids, and PCR products. Constructs involving genes with potential metal-reducing activity and the influenza A virus matrix protein were critical for subsequent expression and functional studies, with correct gene insertion confirmed through sequencing and structural analysis.

### 4.7. Expression, Purification, and Analysis of Proteins with Potential Gold-Reducing Activity

Cultures of *E. coli* BL21(DE3) cells transformed with recombinant vectors (pET101/*cdr*, pET21b/*gor*, pET21b/*oxi*, pET21b/*trmFO*, and pET21b/*trxB*) were grown at 30 °C and 37 °C until reaching an OD_600nm_ of 0.5. Protein expression was induced with 1 mM IPTG (Invitrogen^®^), and samples were taken at various time points to optimize expression conditions. Crude extracts were analyzed by SDS-PAGE to confirm protein expression. Additionally, silver staining [79] was performed to assess the effect of gold nanostructures on influenza A H1N1 virus proteins, specifically examining viral hemagglutinin.

Purification of the expressed proteins was performed using Ni^2+^ affinity chromatography as we described previously [29]. In brief, crude extracts were loaded onto a HisTrap HP column (HisTrap^®^ HP, Cytiva^®^, Marlborough, MA, USA), washed, and eluted with an imidazole-containing buffer. Eluted fractions were quantified and dialyzed, and the purity of the purified proteins was confirmed by SDS-PAGE. These purified proteins were then tested for their potential gold-reducing activity using HAuCl_4_ as a substrate at 25 °C.

### 4.8. Synthesis, Purification, and Characterization of Gold Nanostructures (AuNS)

Gold nanostructures (AuNS) were synthesized in vitro using purified recombinant Gor protein from the environmental isolate BNF01, following Figueroa et al. (2018) [16]. The reaction involved 20 mM Tris-HCl buffer (pH 9.0), 1 mM NADPH, 1 mM HAuCl_4_, and 50 μg of the purified protein. Post-synthesis, AuNS were purified and concentrated using Amicon Ultra-4 3 kDa (Amicon^®^, Darmstadt, Germany) filtration units and centrifuged at 7500× *g* for 10 min. The concentration of gold in the AuNS was measured by flame atomic absorption spectrometry (Analityk Jena Model Anova 350, Jena, Germany).

The size and morphology of AuNS were characterized by transmission electron microscopy (TEM) using a Hitachi HT7700 microscope (CEDENNA, University of Santiago de Chile), whereby a drop of the nanostructures was placed on a copper grid coated with formvar and carbon film and observed at 100 kV. Their chemical composition was determined using energy-dispersive X-ray spectroscopy (EDX) in conjunction with scanning electron microscopy (SEM) on a Zeiss EVO MA10 microscope (Oberkochen, Germany) equipped with an Oxford Instruments X-act system and Penta FET precision detector (Boston, MA, USA). The UV–Vis absorption spectrum was recorded between 400 and 800 nm using a Jasco V-730 spectrophotometer (Tokyo, Japan). Additionally, the size distribution and zeta potential of AuNS were analyzed using a Zetasizer Nano ZS90 (Malvern, UK), with samples sonicated for size measurements and ionic strength adjusted with KCl for zeta potential analysis.

### 4.9. Cultivation, Propagation, and Molecular Analysis of Influenza A H1N1 Virus in MDCK Cells

Madin–Darby canine kidney (MDCK) cells (ATCC/CCL-34, Manassas, VA, USA) were cultured in high-glucose DMEM (Cytiva^®^) supplemented with 10% fetal bovine serum (FBS) (Gibco^®^, Waltham, MA, USA) and penicillin–streptomycin (Corning^®^, New York, NY, USA) at 37 °C in a 5% CO_2_ atmosphere to maintain optimal growth conditions. The cells were subcultured regularly to ensure that they remained healthy and viable for experiments. For the propagation of the influenza A H1N1 virus (A/Virginia/ATCC1/2009 VR-1736), MDCK cells were grown to 80–90% confluence, which is ideal for infection. The cells were then infected with the virus at a multiplicity of infection (MOI) of 0.1. After a 60 min adsorption period to allow the virus to attach to the cells, the medium was replaced with Opti-MEM I (Gibco^®^) supplemented with TPCK-treated trypsin (Thermo Fisher Scientific^®^, Waltham, MA, USA), which facilitates viral entry and replication. The infected cells were incubated for 72 h at 37 °C in a 5% CO_2_ atmosphere. Post-incubation, the viral supernatant was harvested, clarified by centrifugation to remove cell debris, concentrated using Spin-X UF 20 50 kDa filtration units (Corning^®^), and stored at −80 °C for further analysis.

The viral titer was determined using the Reed–Muench method [80]. This involved preparing serial 1:10 dilutions of the viral supernatant, which were then used to infect a monolayer of MDCK cells in 96-well plates. After 48 h, the cells were observed for cytopathic effects, and the endpoint dilution, which indicates the concentration at which 50% of the wells show viral-induced cell death, was used to calculate the viral titer.

RNA was extracted from the viral supernatant using TRIsure reagent (Bioline^®^, Memphis, TN, USA), which effectively lyses the virus and stabilizes RNA. The RNA was then purified through phase separation with chloroform and precipitated with isopropanol. The RNA pellet was washed with ethanol, dried, and resuspended in nuclease-free water. For cDNA synthesis, the purified RNA was first denatured by heating to 70 °C to remove secondary structures. Then, using M-MLV reverse transcriptase (Promega^®^), cDNA was synthesized in a reaction containing specific primers for the influenza A matrix protein gene, ensuring accurate and efficient reverse transcription.

The correct insertion of the influenza A matrix protein gene into the pGEM-T Easy vector was verified by digesting the recombinant plasmid with the *Eco*RI-HF restriction enzyme. The digestion products were analyzed by agarose gel electrophoresis to confirm the presence and size of the insert.

For absolute quantification of viral RNA by RT-qPCR, a calibration curve was generated using serial dilutions of the recombinant pGEM-T Easy/Mseg vector. The RT-qPCR was performed using the Brilliant III Ultra-Fast QRT-PCR Master Mix (Agilent Technologies^®^, Santa Clara, CA, USA) with specific TaqMan probes designed to target the matrix protein gene. The amplification conditions included reverse transcription at 55 °C for 10 min, followed by an initial denaturation at 95 °C and 40 cycles of denaturation and annealing/extension. The viral load in the samples was calculated based on the calibration curve derived from known quantities of the recombinant vector.

Finally, DNA and RNA quantification was performed using a TECAN Infinite M200 Pro plate reader with a NanoQuant plate, which allows for precise measurements even with small sample volumes. The purity of the nucleic acids was assessed by the absorbance ratios at 260/280 nm and 260/230 nm, ensuring the integrity and quality of the samples for downstream applications.

### 4.10. Evaluation of Virucidal Activity, Cytotoxicity, and Hemagglutination Effects of Gold Nanostructures (AuNS)

A hemagglutination assay was performed to determine the hemagglutination titer (HA) of influenza A H1N1 virus using a 96-well U-bottom plate. Serial dilutions of the viral supernatant were mixed with 0.5% human erythrocytes, and the HA titer was defined as the highest dilution showing complete agglutination. To assess the effect of gold nanostructures on hemagglutination, the virus was treated with 100 µg/mL AuNS at various incubation times (0, 10, and 60 min), and the same assay was conducted. The 0 min incubation sample serves as a control to establish the initial viral titer before treatment. Untreated viral supernatant served as a positive control, while 1X PBS and 100 µg/mL AuNS solutions were used as negative controls.

The virucidal activity of biosynthesized AuNS was evaluated both in suspension and on non-porous stainless-steel surfaces following the guidelines of the European Committee for Standardization [17,18]. The influenza A H1N1 virus (10 × 10^6^ TCID_50_/mL) was incubated with 100 µg/mL of AuNS, and viral titers were measured after 10 and 60 min of incubation at 25 °C with shaking. For surface tests, stainless steel disks were used, whereby the virus was adsorbed onto the disks before adding AuNS. After the incubation period (0, 10, and 60 min), samples were processed, and viral titers were determined using the Reed–Muench method [80]. Controls included standardized-hardness water (WSH), a 1:600 dilution of the peroxide-based disinfectant Virkon^®^ as a positive control, and a commercial copper nanostructure biocidal control with a size < 10 nm.

The cytotoxicity of AuNS on MDCK cells was assessed using the MTT assay (3-(4,5-dimethylthiazol-2-yl)-2,5-diphenyl tetrazolium bromide) (Calbiochem^®^, Darmstadt, Germany) as described by Xiang et al. (2011) [81]. Cells were treated with AuNS concentrations ranging from 0 to 800 μg/mL, and cell viability was measured spectrophotometrically at 550 nm after a 24 h incubation period. Additionally, the IC_50_ of AuNS against influenza A H1N1 was determined by incubating various concentrations of AuNS with the virus, followed by viral titer measurement using the Reed–Muench method [80].

### 4.11. Evaluation of Virion Protein Integrity

The integrity of influenza A H1N1 virion proteins was assessed following the protocol by Fujimori et al. (2012) [68]. A mixture of the virus with 100 µg/mL of gold nanostructures was incubated for 60 min at 25 °C with shaking. As a control, AuNS were replaced with standardized-hardness water. After incubation, the mixtures were centrifuged, and the supernatants were quantified by the Bradford method. Proteins were then subjected to SDS-PAGE, followed by silver staining to evaluate protein integrity, and the remaining supernatant was analyzed by transmission electron microscopy (TEM).

### 4.12. Data Graphing and Statistical Analysis

Data graphing and statistical analysis were performed using GraphPad Prism 8.00 software (GraphPad Software Inc., La Jolla, CA, USA). The *t*-test and analysis of variance (ANOVA) were performed with a significance level of *p* < 0.05. Statistical significance was indicated as follows: * for values with *p* < 0.05; ** for values with *p* < 0.01; *** for values with *p* < 0.001; **** for values with *p* < 0.0001; and ns for non-significant differences.

## 5. Conclusions

This study provides valuable insights into the antiviral potential of biosynthesized gold nanostructures (AuNS) against influenza A H1N1 virus. The research successfully identified environmental bacterial isolates with significant resistance to HAuCl_4_, notably BNF01 and MF16, which also demonstrated a strong ability to reduce Au(III). Through in silico analysis, five proteins with potential Au(III) reductase activity were identified, with the recombinant Gor protein from BNF01 showing high reductive capacity and enabling the biosynthesis of AuNS.

The virucidal assays revealed that these biosynthesized AuNS effectively reduced viral titers in suspension assays, confirming their potential as antiviral agents. The observed reduction in viral titers, particularly in suspension assays, was accompanied by evidence of disruption to viral proteins, as demonstrated by hemagglutination assays, TEM imaging, and SDS-PAGE analysis. Despite these promising results, the AuNS did not show the same level of efficacy on non-porous stainless-steel surfaces, highlighting the need for further investigation into the conditions that influence their antiviral activity.

The findings suggest that the antiviral mechanism of AuNS likely involves the interaction with and degradation of viral surface proteins, impairing the virus’s ability to infect host cells. This study underscores the potential of AuNS as a novel antiviral material, but it also indicates that additional research is necessary to fully understand their mechanism of action and to optimize their use in different environments, particularly for surface disinfection.

In conclusion, the biosynthesized AuNS present a promising avenue for antiviral applications, particularly in suspension environments. Future studies should focus on refining the synthesis process, further characterizing the interaction of AuNS with viral components, and exploring their effectiveness against a broader range of viruses and in various practical settings.

## Figures and Tables

**Figure 1 ijms-26-01934-f001:**
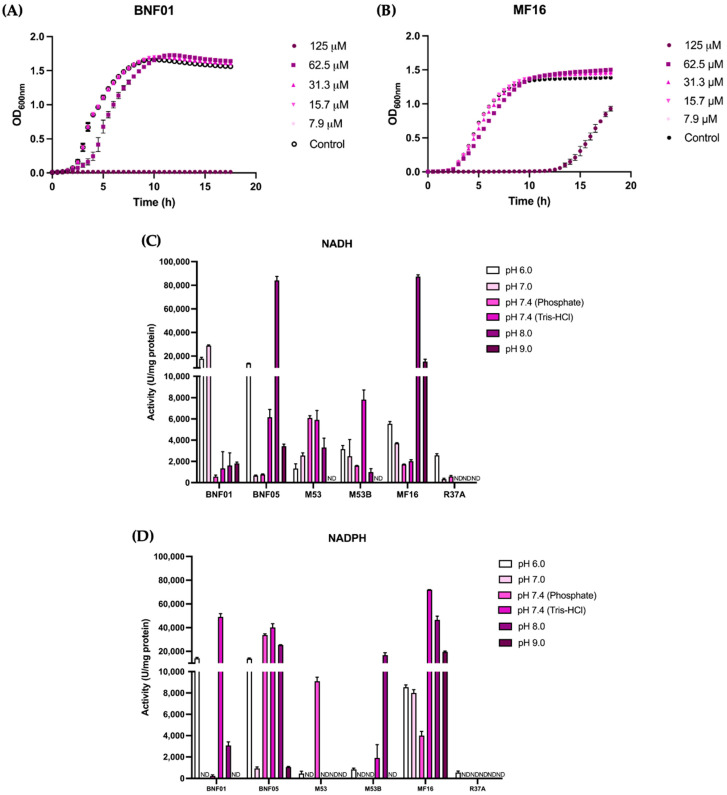
Evaluation of the gold resistance performance of environmental isolates. Growth curves of the environmental isolates BNF01 (**A**) and MF16 (**B**) resistant to Au(III) exposed to sublethal concentrations of HAuCl_4_ (based on the MIC). The environmental isolates were grown in LB medium in the absence (black circles) or presence of HAuCl_4_ (concentration indicated next to each graph). The curves shown represents the environmental isolates most resistant to gold. Each point represents the mean of three independent experiments ± standard deviation. (**C**,**D**) Gold-reducing activity mediated by crude extracts from environmental isolates resistant to Au(III). The measurement of gold-reducing activity with HAuCl_4_ was performed in the presence of NADH (**C**) and NADPH (**D**) cofactors across different pH levels (colored bars). “ND” indicates no reducing activity observed under the tested conditions. The bars represent the mean of three independent experiments ± standard deviation.

**Figure 2 ijms-26-01934-f002:**
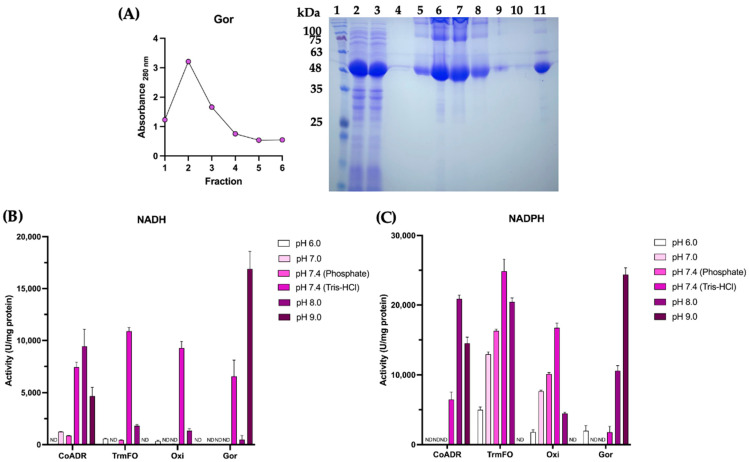
Recombinant proteins from environmental isolates display gold-reducing activity. (**A**) Protein purification process for Gor with potential Au(III) reductase activity. Left: Elution profile of Gor protein. Right: SDS-PAGE analysis. Lane 1: Molecular weight marker. Lane 2: Crude extract before column purification. Lane 3: Crude extract after column purification. Lane 4: Column wash. Lanes 5–10: Eluted protein fractions. Lane 11: Purified protein after dialysis. (**B**,**C**) Au(III) reductase activity mediated by purified proteins. The measurement of HAuCl_4_ reductase activity was performed with 50 µg of purified protein in the presence of NADH (**B**) and NADPH (**C**) cofactors across different pH levels (colored bars). “ND” indicates no reductase activity observed under the tested conditions. Bars represent the mean of three independent experiments ± standard deviation.

**Figure 3 ijms-26-01934-f003:**
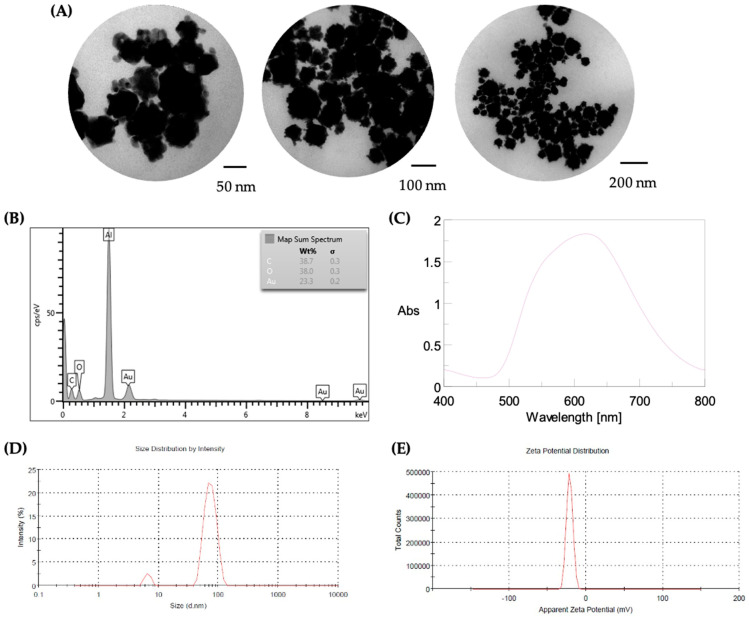
Characterization of gold nanostructures (AuNS) biosynthesized in vitro by purified recombinant Gor protein. (**A**) TEM micrographs show urchin-like structures resulting from gold reduction. (**B**) SEM–EDX analysis indicates that the primary composition of the biosynthesized AuNS is carbon, oxygen, and gold. (**C**) UV–Vis absorption spectrum shows a maximum absorbance at 622 nm. (**D**) Size distribution analysis by DLS shows two predominant distributions in the biosynthesized gold nanostructures. (**E**) Zeta potential distribution determined for the biosynthesized gold nanostructures was −21.4 mV.

**Figure 4 ijms-26-01934-f004:**
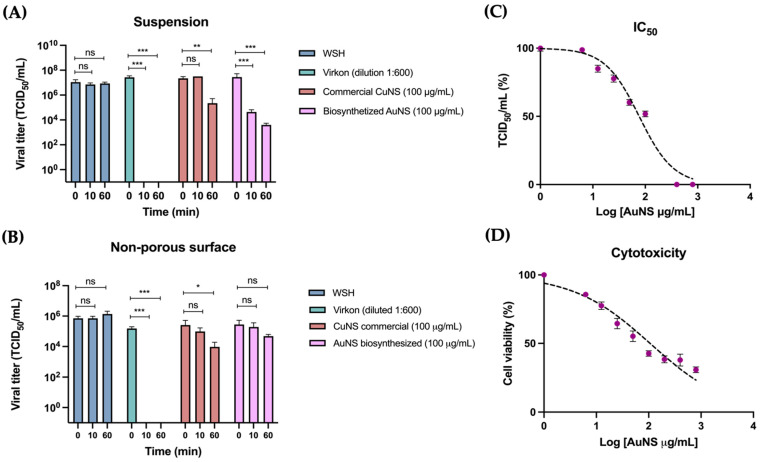
Virucidal activity of AuNS. (**A**,**B**) Virucidal assays against influenza A virus in suspension and on non-porous stainless-steel surfaces. Influenza A virus inactivation was evaluated in suspension (**A**) and on non-porous stainless-steel surfaces (**B**) using standardized-hardness water (WSH), Virkon^®^ solution (1:600 dilution), and commercial copper nanostructures (CuNS) and biosynthesized gold nanostructures (AuNS) with short (10 min) and long (60 min) contact times. The Reed–Muench quantification method was used to determine the TCID_50_/mL. Results represent three independent replicates ± standard deviation. (**C**) Inhibition of cytopathic effect caused by influenza A H1N1 virus after treatment with different concentrations of AuNS. R^2^ = 0.9722. Each point represents the mean of three independent experiments ± standard deviation. (**D**) Cytotoxicity of AuNS in MDCK cell cultures after 48 h of incubation, determined by the MTT assay. Non-linear regression curves (dashed line) were generated with concentrations expressed on a logarithmic scale. R^2^ = 0.9724. Each point represents the mean of three independent experiments ± standard deviation. * *p* < 0.05; ** *p* < 0.01; *** *p* < 0.001; ns, not significant.

**Figure 5 ijms-26-01934-f005:**
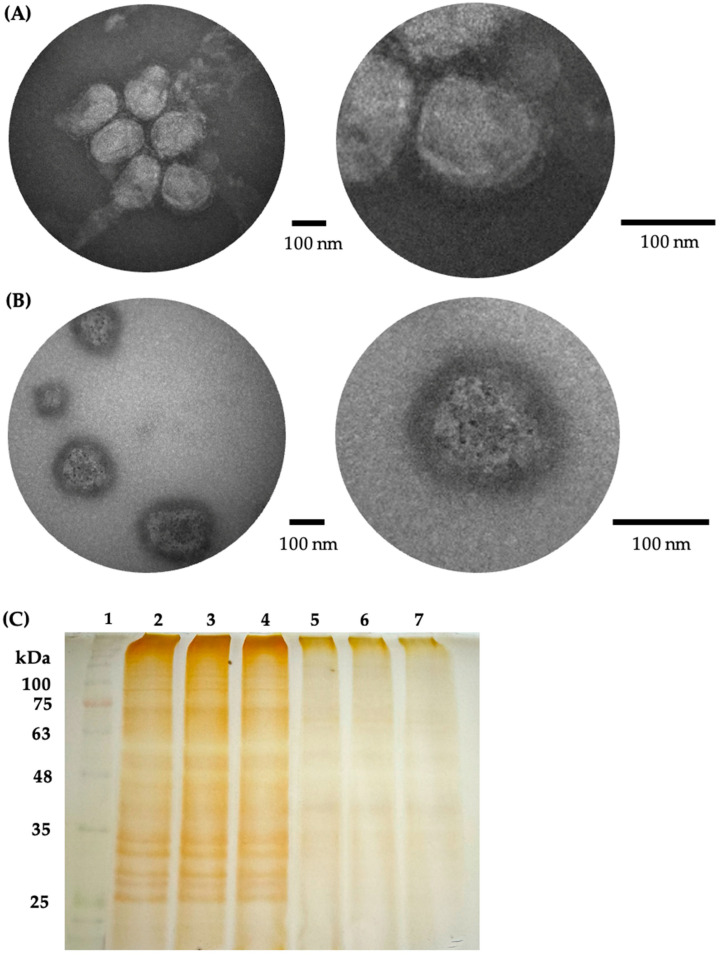
Ultrastructure characterization of the AuNS and viral degradative activity. (**A**,**B**) Evaluation of influenza A H1N1 virus integrity by transmission electron microscopy (TEM) post-treatment with biosynthesized AuNS. Enriched influenza A H1N1 virus was incubated with standardized-hardness water (WSH) (**A**) or 100 µg/mL biosynthesized AuNS (**B**) for 60 min with constant agitation at 25 °C and subsequently visualized by TEM. (**C**) Degradative effects of biosynthesized AuNS treatment on influenza A H1N1 virus visualized by silver-stained SDS-PAGE. Lane 1: Molecular weight marker (AccuRuler RGB PLUS Prestained Protein Ladder). Lanes 2–4: Triplicate aliquots of influenza A H1N1 virus treated with WSH for 60 min. Lanes 5–7: Triplicate aliquots of influenza A H1N1 virus treated with 100 µg/mL of AuNS for 60 min.

**Table 1 ijms-26-01934-t001:** Minimum inhibitory concentrations (MIC), minimum bactericidal concentrations (MBC), and growth parameters of environmental isolates in the presence of HAuCl_4_.

Isolate	MIC (mM)	MBC (mM)	OD_600nm_ max (%)	Growth Rate (%)	Generation Time (%)
MF01	0.125	0.125	WOV	WOV	WOV
MF02	0.125	0.25	ND	ND	ND
MF07	0.125	0.25	ND	ND	ND
MF10	0.125	0.125	ND	ND	ND
MF15	0.125	0.125	ND	ND	ND
MF16	0.25	0.25	WOV	WOV	WOV
MF17	0.125	0.25	ND	ND	ND
MF18	0.125	0.125	ND	ND	ND
MF19	0.125	0.125	ND	ND	ND
MF20	0.125	0.125	ND	ND	ND
BNF01	0.25	0.25	WOV	↓ 44	↑ 78.9
BNF05	0.25	0.25	WOV	↓ 28.8	↑ 40.9
BNF20	0.125	0.25	ND	ND	ND
BNF22	0.03125	0.25	ND	ND	ND
M53	0.25	0.5	↓ 16.1	↓ 32.5	WOV
M53B	0.0625	0.5	↓ 10.6	↓ 40.3	WOV
R37A	0.25	0.5	↓ 13.3	WOV	↓ 13.3
BW 25113	0.125	0.25	ND	ND	ND

This table presents a comprehensive summary of the resistance of various environmental bacterial isolates to HAuCl_4_. The MIC and MBC values are shown alongside the main variations in bacterial growth parameters, including maximum optical density at 600 nm (OD_600nm_), growth rate, and generation time. The MIC and MBC values were determined based on six independent measurements, while the growth parameters were evaluated in the presence of sublethal concentrations of HAuCl_4_. Isolates that exhibited the highest resistance, as indicated by the MIC and MBC values, and significant variations in growth parameters are highlighted. The arrows ‘↓’ and ‘↑’ indicate decreases or increases in the parameters with respect to the control condition (no treatment with HAuCl_4_), respectively. ‘WOV’ indicates ‘without variation’, meaning no significant change in the measured parameter compared to the control condition. ‘ND’ stands for ‘not determined’, indicating that the data are not available or not applicable.

**Table 2 ijms-26-01934-t002:** In silico identification of proteins with probable Au(III) reductase activity in the environmental isolate BNF01.

Protein Name	Gene Name	Gene Size (bp)
Thioredoxin reductase	*trxB*	942
Glutathione reductase	*gor*	1524
L-amino acid oxidase	*oxi*	724
Methylenetetrahydrofolate-tRNA-(uracil-5)-methyltransferase (TrmFO)	*trmFO*	1308
Coenzyme A disulfide reductase	*cdr*	1323

**Table 3 ijms-26-01934-t003:** Initial and final viral titers obtained in virucidal assays in suspension and on non-porous stainless-steel surfaces.

Sample	Viral Title: Suspension			Viral Title: Non-Porous Surfaces		
	0 min	10 min	60 min	0 min	10 min	60 min
**WSH**	1.1 × 10^7^	7.1 × 10^6^	8.5 × 10^6^	7.1 × 10^5^	7.1 × 10^5^	1.4 × 10^6^
**Virkon^®^**	2.7 × 10^7^	NE	NE	4.5 × 10^5^	NE	NE
**Commercial CuNS**	2.3 × 10^7^	3.2 × 10^7^	2.2 × 10^5^	8.5 × 10^5^	9.7 × 10^4^	9.4 × 10^3^
**Biosynthesized AuNS**	2.8 × 10^7^	4.3 × 10^4^	4.0 × 10^3^	2.4 × 10^5^	1.9 x 10^5^	4.8 × 10^4^

Viral titers were determined using the Reed–Muench quantification method to obtain TCID_50_/mL after exposing the virus to the indicated compound for short (10 min) and long (60 min) contact times. NE: No cytopathic effect observed in any of the evaluated virus dilutions.

## Data Availability

All data generated or analyzed during this study are included in this published article and the Appendix A.

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
