# Peer review of "Biosynthesis of Gold Nanostructures and Their Virucidal Activity Against Influenza A Virus"

_ijms, 2025, doi:10.3390/ijms26051934_

Round 1
Reviewer 1 Report
Comments and Suggestions for Authors
The manuscript on “Biosynthesis of gold nanostructures ….virucidal activity against Influenza A virus” describes the biosynthesis and anti-viral effect against Influenza A virus. The article is interesting and addressed the virucidal activity of their biosynthesized Au NS. The article is suitable for publication with major revision following as
1. The portion covering the biosynthesis of Au NS looks complex and overloaded as extraction or list of 18 bacterial isolates, their MIC, MBC and then narrow down further to identify protein with Au reductase activity. In my opinion, the authors should reduce the content and become specific of chosen bacterial isolation and protein for Au biosynthesis.
2. SEM EDX confirmed the presence of C, O from bioactive agents (Gor protein). The authors should include the FTIR spectra to further support the involvement of Gor protein in the synthesis and stabilization of Au NS.
3. TEM images do not show ultrafine (<10 nm) Au NS. Even the smallest NPs on TEM images are bigger than 15 nm. Whereas DLS shows smaller size of Au NS which is approx. 6 nm. It is difficult to believe that these two measurements (TEM, DLS) are performed using the same sample. The aggregation of Au NS which is visible in TEM images is not reflected in DLS measurement.
4. The authors made a good attempt on the understanding of the mechanism behind the antiviral activity of biosynthesized Au NS through electron microscopy and SDS page analysis. Please cite a relevant review article (https://pubs.acs.org/doi/10.1021/acsptsci.2c00195) on Nanoscale interaction mechanism of antiviral activity.
5. Why was plaque assay not performed which is a gold standard assay to measure TCID50/mL ?
6. Why authors choose 100 µg/mL concentration of Au NS for virucidal experiments which gave 50 % cytotoxicity on MDCK cell culture. However, low or negligible cytotoxicity is one of the most important requirements for any potential antiviral agent.
7. The authors added sufficient regerences on biosynthesis of Au NS but not previous article on their antiviral effects. Please add previous relevant references on antiviral activity of gold nanoparticles examples (doi: 10.3390/v11121111, 10.3390/molecules26195960, https://doi.org/10.1016/j.biopha.2025.117844).
Author Response
Comments and Suggestions for Authors
The manuscript on “Biosynthesis of gold nanostructures ….virucidal activity against Influenza A virus” describes the biosynthesis and anti-viral effect against Influenza A virus. The article is interesting and addressed the virucidal activity of their biosynthesized Au NS. The article is suitable for publication with major revision following as:
- The portion covering the biosynthesis of Au NS looks complex and overloaded as extraction or list of 18 bacterial isolates, their MIC, MBC and then narrow down further to identify protein with Au reductase activity. In my opinion, the authors should reduce the content and become specific of chosen bacterial isolation and protein for Au biosynthesis.
Response: We appreciate the reviewer’s suggestion regarding the complexity of the AuNS biosynthesis section. This study was designed as an exploratory approach, starting from a broad selection of 18 environmental bacterial isolates and multiple candidate proteins to systematically identify the most efficient strains and key enzymatic components involved in Au(III) reduction. This extensive screening was essential to ensure the reliability of the results and to provide a comprehensive understanding of the bacterial mechanisms leading to gold nanostructure biosynthesis.
To enhance clarity and readability, we have refined the description of bacterial isolates, emphasizing the six with the highest performance in gold resistance and reduction capacity. Similarly, while multiple proteins exhibited potential Au(III) reductase activity, we have highlighted the Gor protein as the primary candidate based on its superior enzymatic efficiency. However, reducing the initial dataset further would compromise the exploratory nature of the study and limit the understanding of the selection process that ultimately led to AuNS biosynthesis.
In addition, throughout the manuscript, we have modified explanations and restructured certain phrases to better convey the rationale behind the selection of bacterial strains and proteins. These modifications improve the flow of information and ensure that the transition from the broader screening to the selected candidates is clearer and more intuitive.
By balancing a comprehensive initial screening with a refined selection of key isolates and proteins, we aimed to maintain the scientific rigor and exploratory depth of the study while ensuring that the results remain focused, interpretable, and accessible to the reader
- SEM EDX confirmed the presence of C, O from bioactive agents (Gor protein). The authors should include the FTIR spectra to further support the involvement of Gor protein in the synthesis and stabilization of Au NS.
Response: We appreciate the reviewer's suggestion. While FTIR spectroscopy was not performed in this study, we provide alternative evidence supporting the role of Gor protein in the synthesis and stabilization of AuNS. i) SEM-EDS Analysis: The detection of carbon and oxygen signals in SEM-EDS analysis indicates the presence of organic biomolecules, likely amino acids and functional groups from the Gor protein, which remained attached to the nanoparticles. ii) Biological Mechanism of AuNS Formation: The synthesis was conducted using a recombinant purified Gor protein, whose enzymatic activity led to the controlled reduction of Au(III) and nanoparticle formation. The ability of this protein to mediate metal reduction and stabilization aligns with previous reports on enzymatic nanoparticle synthesis. iii) Zeta Potential Measurements: The observed negative zeta potential of AuNS suggests the presence of surface biomolecules, which contribute to nanoparticle stabilization by electrostatic repulsion. Iv) Supporting Literature: Previous studies have shown that enzymatic reduction and stabilization of metal nanoparticles occur via protein-nanoparticle interactions, particularly involving thiol (-SH), amine (-NH₂), and carboxyl (-COOH) groups. These interactions are consistent with those expected from Gor protein.
Thus, while FTIR data are not available, the combination of SEM-EDS results, the biological synthesis approach, zeta potential measurements, and supporting literature strongly supports the role of Gor protein in the synthesis and stabilization of AuNS.
- TEM images do not show ultrafine (<10 nm) Au NS. Even the smallest NPs on TEM images are bigger than 15 nm. Whereas DLS shows smaller size of Au NS which is approx. 6 nm. It is difficult to believe that these two measurements (TEM, DLS) are performed using the same sample. The aggregation of Au NS which is visible in TEM images is not reflected in DLS measurement.
Response: The reviewer is correct in identifying this discrepancy. However, both measurements were indeed conducted on the same sample. The differences arise due to sample preparation: i) For TEM imaging, multiple droplets were deposited onto the copper grid, leading to increased local concentrations and nanoparticle aggregation due to their zeta potential; ii) For DLS analysis, the sample was sonicated before measurement, preventing aggregation and reflecting the true size distribution of individual particles. This explanation has now been added to the manuscript (new lines 674-679).
- The authors made a good attempt on the understanding of the mechanism behind the antiviral activity of biosynthesized Au NS through electron microscopy and SDS page analysis. Please cite a relevant review article (https://pubs.acs.org/doi/10.1021/acsptsci.2c00195) on Nanoscale interaction mechanism of antiviral activity.
Response: We appreciate this valuable suggestion. The recommended reference has been incorporated into the revised discussion to strengthen the analysis of the possible nanoscale interactions responsible for the virucidal activity of AuNS. It is important to emphasize that virucidal activity is not the same as antiviral activity. Antiviral activity refers to interfering with one of the replication cycle stages of the virus under study. In our case, we evaluated the virucidal action of AuNS against Influenza A virus, which is related to the inactivation of viral particles through mechanisms that include: i) Degradation of viral surface proteins; ii) Damage to viral envelopes and capsids; and iii) Oxidation of viral particles. Regarding this, the most relevant studies supporting our results and the possible mechanism behind AuNS activity have been cited in the discussion (new lines 813-814, 821-823, 838-841; references 57, 58, 68). The primary mechanism proposed is protein degradation on the viral surface, leading to virus inactivation.
- Why was plaque assay not performed which is a gold standard assay to measure TCID50/mL ?
Response: We used TCID50/mL following the standardized virucidal evaluation protocol established by the European Committee for Standardization (CEN), which allows for either plaque assays or TCID50 measurements to determine viral titers in virucidal assays on suspensions and non-porous surfaces (references 17 and 18). Additionally, given the multiple treatments and contact times evaluated, TCID50/mL was the more efficient method, requiring fewer resources and a shorter execution time compared to plaque assays.
- Why authors choose 100 µg/mL concentration of Au NS for virucidal experiments which gave 50 % cytotoxicity on MDCK cell culture. However, low or negligible cytotoxicity is one of the most important requirements for any potential antiviral agent.
Response: The reviewer is correct in noting that cytotoxicity is a critical factor for antiviral agents. However, it is important to distinguish between virucidal and antiviral activity: i) Antiviral agents must be non-toxic as they are used in therapeutic applications; ii) Virucidal agents, such as disinfectants and surface protectants, have different requirements, where cytotoxicity is not as critical. We selected 100 µg/mL as the lowest concentration that resulted in a significant reduction of initial viral titers in both suspension and non-porous surface assays.
- The authors added sufficient regerences on biosynthesis of Au NS but not previous article on their antiviral effects. Please add previous relevant references on antiviral activity of gold nanoparticles examples (doi: 10.3390/v11121111, 10.3390/molecules26195960, https://doi.org/10.1016/j.biopha.2025.117844).
Response: Done. We appreciate this recommendation. The suggested references have been added to the manuscript to provide a broader context on the antiviral potential of gold nanoparticles.
It is important to note that this study specifically evaluates the virucidal effect of biosynthesized AuNS against Influenza A virus. Unlike antiviral compounds, which interfere with viral replication, virucidal agents work by directly inactivating viral particles.
There are few studies available on the virucidal effects of gold nanoparticles, but the most relevant studies published to date have been cited and included in the discussion of our results (references 57 and 58). These references provide valuable insights into how nanostructures interact with viruses and support our findings on the mechanisms of AuNS-mediated virus inactivation.
Reviewer 2 Report
Comments and Suggestions for Authors
The manuscript involving the biosynthesis of gold nanoparticles and the activity of these nanostructures against selected viruses has been reviewed.
It is necessary that the introduction subsection be corrected because this part of the manuscript should refer directly to the conducted research and should show the state of knowledge in this field. The introduction should justify the undertaking of the research by the Authors.
The Authors show that the "wild" strains were isolated from the environment, but the procedures (including molecular biology methods) to confirm the species affiliation of the isolates were not shown. It should also be considered to deposit these strains in an appropriate collection of microorganisms.
It must also be emphasized that in the experiments conducted it was not proven that the synthesis of gold nanoparticles can be performed by NADH or NADPH (without cell extracts).
The obtained gold nanoparticles show aggregation, which is probably the result of “removing” proteins from the colloidal solution, which are stabilizers of these nanostructures. Explain what biological molecules are stabilizers of the obtained gold nanoparticles.
The Authors showed that gold nanoparticles show moderate antiviral activity. The antiviral activity of gold nanoparticles has already been shown earlier and is not a real scientific novelty.
Author Response
The manuscript involving the biosynthesis of gold nanoparticles and the activity of these nanostructures against selected viruses has been reviewed.
It is necessary that the introduction subsection be corrected because this part of the manuscript should refer directly to the conducted research and should show the state of knowledge in this field. The introduction should justify the undertaking of the research by the Authors.
Response: Done. We have revised the introduction to more clearly justify the relevance of biosynthesized AuNS as a sustainable alternative to chemically synthesized nanostructures and their potential as virucidal agents against Influenza A virus.
The Authors show that the "wild" strains were isolated from the environment, but the procedures (including molecular biology methods) to confirm the species affiliation of the isolates were not shown. It should also be considered to deposit these strains in an appropriate collection of microorganisms.
Response: Done. We confirm that 16S rRNA sequencing was performed for bacterial identification. This detail has now been explicitly mentioned in the manuscript (new lines 130-135).
It must also be emphasized that in the experiments conducted it was not proven that the synthesis of gold nanoparticles can be performed by NADH or NADPH (without cell extracts).
Response: We appreciate the reviewer's observation. The current study specifically focuses on protein-mediated AuNS biosynthesis, where the purified Gor protein acts as the primary reducing and stabilizing agent. We did not evaluate the direct role of NADH/NADPH alone in AuNS synthesis, as the emphasis was on enzymatic nanoparticle formation rather than abiotic reduction. However, controls of the nanoparticle synthesis reaction were performed without the Gor protein, conditions under which no abiotic AuNS formation was obtained. Despite this, we acknowledge that NADH and NADPH may contribute to metal reduction under certain conditions, and this remains an interesting aspect for future investigations. We have now included this point in the discussion section to highlight potential follow-up studies (new lines 638-644).
The obtained gold nanoparticles show aggregation, which is probably the result of “removing” proteins from the colloidal solution, which are stabilizers of these nanostructures. Explain what biological molecules are stabilizers of the obtained gold nanoparticles.
Response: Done. As described in the manuscript, the presence of carbon and oxygen detected via SEM-EDS is associated with stabilizing agents, which likely correspond to amino acids and other biomolecules from the Gor protein that were not completely removed during the purification process. These biomolecules play a crucial role in stabilizing the biosynthesized AuNS, as they are involved in the mechanism of nanoparticle synthesis mediated by the Gor protein. This relationship is discussed in detail in the manuscript (new lines 663-666) and supported by reference 41.
The Authors showed that gold nanoparticles show moderate antiviral activity. The antiviral activity of gold nanoparticles has already been shown earlier and is not a real scientific novelty.
Response: It is important to distinguish between antiviral and virucidal activity. Antiviral activity involves interfering with specific stages of the viral replication cycle, whereas virucidal activity refers to the direct inactivation of viral particles. In our study, we evaluated the virucidal action of biosynthesized AuNS against Influenza A virus, demonstrating that AuNS cause degradation of viral surface proteins, damage to viral envelopes and capsids, and oxidative modifications leading to virus inactivation. The discussion includes key references (new lines -814, 821-823, 838-841; references 57, 58, 68) that support our results and the proposed mechanism behind AuNS virucidal action. Additionally, the available literature primarily focuses on chemically synthesized gold nanoparticles, which differ significantly in composition, surface chemistry, and biological interactions compared to our biosynthesized AuNS. This distinction is now explicitly addressed in the revised manuscript.
Reviewer 3 Report
Comments and Suggestions for Authors
In the manuscript submitted to me for review entitled "Biosynthesis of Gold Nanostructures and Their Virucidal Activity Against Influenza A Virus“ the authors Fernanda Contreras, Katherine Rivero, Jaime Andrés Rivas-Pardo, Fabiana Andrea Liendo, Rodrigo Segura, Nicole Neira, Mauricio Arenas-Salinas, Marcelo Cortez-San Martín and Felipe Arenas present a study in which the biosynthesis of gold nanostructures (AuNS) from environmental bacteria is investigated. The virucidal potential of these AuNS against Influenza A virus was also determined.
My remarks and recommendations to the authors are:
1. Table 1 does not indicate what the arrows mean. Let's add an explanation.
2. Figure 4 is not sized to fit the page size and is not visible in its entirety. Let's reduce the size of the figure so that it can be seen in its entirety.
3. Where was the culture of E.coli BL21 (DE3) cells purchased?
4. It is not indicated where the MDCK cells were purchased.
5. In many places in the Materials and Methods section, the manufacturer with the city and country of the consumables used is missing. Somewhere the manufacturer is indicated, but without the city and country. Let's review the text and supplement the information.
6. Has the influenza virus also been cultured in chicken embryos? It is not indicated in the text.
7. On line 1032 it is stated that:
" the virus was treated with 100 µg/mL AuNS at various incubation times (0, 10, and 60 minutes)"
Does this mean that at incubation time 0 min, no incubation actually occurs and the sample is a viral control?
8. On lines 1040-1041 it is not stated after how many hours of incubation the viral titers are determined.
9. On lines 1041 and 1051 it is mentioned that the Reed and Muench method is used, but no citation is given. This is a basic methodology first described in a publication in 1938. I recommend that this citation be included.
Author Response
In the manuscript submitted to me for review entitled "Biosynthesis of Gold Nanostructures and Their Virucidal Activity Against Influenza A Virus“ the authors Fernanda Contreras, Katherine Rivero, Jaime Andrés Rivas-Pardo, Fabiana Andrea Liendo, Rodrigo Segura, Nicole Neira, Mauricio Arenas-Salinas, Marcelo Cortez-San Martín and Felipe Arenas present a study in which the biosynthesis of gold nanostructures (AuNS) from environmental bacteria is investigated. The virucidal potential of these AuNS against Influenza A virus was also determined.
My remarks and recommendations to the authors are:
- Table 1 does not indicate what the arrows mean. Let's add an explanation.
Response: Done. An explanation for the arrows in Table 1 has been added (new lines 119-121).
- Figure 4 is not sized to fit the page size and is not visible in its entirety. Let's reduce the size of the figure so that it can be seen in its entirety.
Response: Done. Figure 4 has been resized to ensure it is fully visible.
- Where was the culture of E.coli BL21 (DE3) cells purchased?
Response: E. coli BL21 (DE3) cells were obtained from Thermo Fisher Scientific (Invitrogen) (new line 865).
- It is not indicated where the MDCK cells were purchased.
Response: Done. MDCK (CCL-34) cells were obtained from ATCC (new line 981).
- In many places in the Materials and Methods section, the manufacturer with the city and country of the consumables used is missing. Somewhere the manufacturer is indicated, but without the city and country. Let's review the text and supplement the information.
Response: Done. We have revised the Materials and Methods section to include the manufacturer details along with the city and country for all consumables used.
- Has the influenza virus also been cultured in chicken embryos? It is not indicated in the text.
Response: The influenza A virus was propagated exclusively in MDCK cells and was not cultured in chicken embryos.
- On line 1032 it is stated that: " the virus was treated with 100 µg/mL AuNS at various incubation times (0, 10, and 60 minutes)" Does this mean that at incubation time 0 min, no incubation actually occurs and the sample is a viral control?
Response: The reviewer is correct. The 0-minute incubation sample serves as a control to establish the initial viral titer before treatment. This clarification has been included in the manuscript (new lines 1032-1034).
- On lines 1040-1041 it is not stated after how many hours of incubation the viral titers are determined.
Response: Done. The manuscript has been updated to specify that viral titers were determined at 0, 10, and 60 minutes (new lines 1041-1043).
- On lines 1041 and 1051 it is mentioned that the Reed and Muench method is used, but no citation is given. This is a basic methodology first described in a publication in 1938. I recommend that this citation be included.
Response: The original Reed & Muench (1938) reference has been added (new lines 995, 1043, 1052).
Round 2
Reviewer 1 Report
Comments and Suggestions for Authors
The authors excellently addressed my concerns and updated the manuscript accordingly. The manuscript is suitable to publish in IJMS.
Author Response
We sincerely appreciate the reviewer’s time and effort in evaluating our manuscript. We are grateful for the constructive feedback, which has significantly contributed to improving the clarity and quality of our work.
We are pleased to hear that our revisions have satisfactorily addressed the concerns raised and that the manuscript is now suitable for publication in IJMS. Thank you for your insightful comments and for helping us refine our study.
Reviewer 2 Report
Comments and Suggestions for Authors I thank the authors for their corrections and improvements to this manuscript.Author Response
We sincerely appreciate the reviewer’s time and effort in evaluating our manuscript. We are grateful for the constructive feedback, which has significantly contributed to improving the clarity and quality of our work.
We are pleased to hear that our revisions have satisfactorily addressed the concerns raised and that the manuscript is now suitable for publication in IJMS. Thank you for your insightful comments and for helping us refine our study